# MM-R³: On (In-)Consistency of Multi-modal Large Language Models (MLLMs)

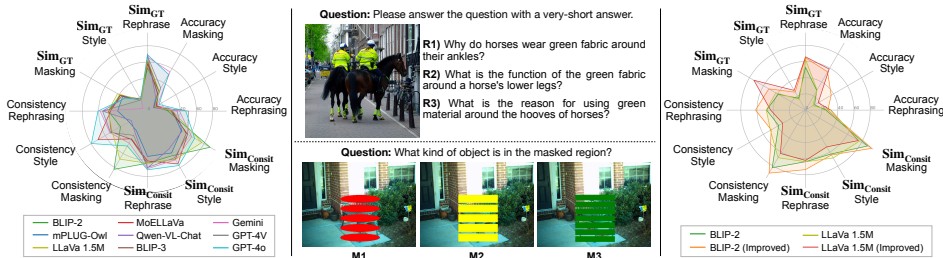

Figure 1: (Left) Overall results of MLLMs on the MM-R³ Benchmark. (Mid) Consider answering the three semantically identical questions for the top image and a given visual abductive reasoning question for the bottom images from the proposed MM-R³ Benchmark. Humans are accurate and consistent in these tasks while MLLMs are much less so. (Right) Results with the proposed adapter.

## ABSTRACT

With the advent of Large Language Models (LLMs) and Multimodal (Visio-lingual) LLMs, a flurry of research has emerged, analyzing the performance of such models across a diverse array of tasks. While most studies focus on evaluating the capabilities of state-of-the-art (SoTA) MLLM models through task accuracy (*e.g.*, Visual Question Answering, grounding) across various datasets, our work explores the related but complementary aspect of *consistency* – the ability of an MLLM model to produce semantically similar or identical responses to semantically similar queries. We note that consistency is a fundamental prerequisite (necessary but not sufficient condition) for robustness and trust in MLLMs. Humans, in particular, are known to be highly consistent (even if not always accurate) in their responses, and consistency is inherently expected from AI systems. Armed with this perspective, we propose the *MM-R³ benchmark*, which analyses the performance in terms of consistency and accuracy in SoTA MLLMs with three tasks: Question Rephrasing, Image Restyling, and Context Reasoning. Our analysis reveals that consistency does not always align with accuracy, indicating that models with higher accuracy are not necessarily more consistent, and vice versa. Furthermore, we propose a simple yet effective mitigation strategy in the form of an adapter module trained to minimize inconsistency across prompts. With our proposed strategy, we are able to achieve absolute improvements of $5.7\%$ and $12.5\%$, on average on widely used MLLMs such as BLIP-2 and LLaVa 1.5M in terms of consistency over their existing counterparts.

## 1 INTRODUCTION

Multimodal Large Language Models (MLLMs) (Liu et al., 2023a; Li et al., 2023b; OpenAI, 2023; Xue et al., 2024), following and often built on top of purely lingual LLM (Brown et al., 2020; Touvron et al., 2023), have recently emerged as incredible tools for a broad range of visual understanding tasks, spanning captioning (Lin et al., 2014; Sharma et al., 2018; Chen et al., 2015), language grounding (Yu et al., 2016; Kazemzadeh et al., 2014; Liu et al., 2019), visual question answering (VQA) (Antol et al., 2015), and many others. As the number of such models and their capabilities explode, the research community is progressively focusing on benchmarking their capabilities by developing new benchmarks and testing harnesses. Notable examples include MM-Bench (Liu et al., 2023c), SEED-Bench (Li et al., 2023a), MM-Vet (Yu et al., 2023), and others that define numerous tasks that capture a broad range of capabilities of such models (*e.g.*, instance counting (Fu et al., 2023), spatial relation understanding (Yu et al., 2023), abductive (Hessel et al., 2022) and deduc-

tive (Park et al., 2020) reasoning, meme comprehension (Li et al., 2023a), *etc*). These benchmarks continue to shed light on the abilities and limitations of MLLMs by analyzing their *accuracy*.

However, despite significant progress in the analyses of MLLM models, *consistency*, the ability to produce identical or semantically equivalent outputs with the same semantic content inputs, remains broadly overlooked. This is a fundamental requirement for MLLMs to be reliably deployable for most tasks. Anecdotally, LLMs and, by extension, MLLMs are sensitive to their prompts which led to the widespread practice of prompt engineering. This is problematic as the models' outputs may vary with the phrasing of a query rather than its actual intent, which undermines their reliability. Consider the example illustrated in Figure 1 (Mid) top: Most humans would realize that while the three questions (i.e R1, R2, and R3) are superficially different, the semantic meaning is the same. Hence even when the correct answer may not perhaps be apparent (*i.e.*, "to be visible"), the same (consistent) answer should be produced. In contrast, asking models like BLIP-2 (Li et al., 2023b) to answer these questions results in varied responses "to protect them from splinters","to protect the horse's legs","to make the hooves more visible" for the three questions considered. Similarly, it is obvious to humans that the object being masked in Figure 1 (Mid) bottom is the same irrespective of the type of the mask, and that the object in question is a "bench" However, Qwen-VL-Chat's responses vary, indicating "a bench","a yellow object", "a green wooden slat sign" for different masks, highlighting the inconsistency in visual modifications.

It may be tempting to equate *accuracy* and *consistency*, but the relationship is more intricate. While it is true that for objective visual tasks (*e.g.*, what color is an object), high accuracy will result in high(er) consistency, current MLLM models are far from this high accuracy regime. Further, for more subjective visual tasks (*e.g.*, abductive reasoning), high accuracy which tends to be measured as being similar to one of the sets of answers, may not lead to high consistency. In general, one can think of consistency as a necessary but not sufficient property of an AI system and one should seek to maximize both *consistency* and *accuracy*.

In this work, we first present a comprehensive analysis of SoTA MLLM models in terms of their consistency. We do so by developing MM-R$^3$ benchmark atop of the VQA task, where we produce both lingual rephrasings of the original questions (by leveraging GPT-3.5) and visual rephrasings of the image (through stylization) and measure both accuracy and consistency of the produced responses. We find that SoTA MLLMs while often quite competitive in accuracy can differ substantially in their consistency of responses. For example, mPLUG-Owl2 (Ye et al., 2024) is much more susceptible to inconsistency when image inputs are perturbed while MoE-LLaVa (Lin et al., 2024) is more consistent in the change of the visual domain than the lingual domain. In addition, we also define an abductive task of predicting the contents of the masked region, where we find models like BLIP-2 and LLaVa 1.5M (Liu et al., 2023a; 2024) are lower in accuracy but have higher consistency. Overall results for MLLMs are illustrated in Figure 1 (Left). We believe these findings both benchmark the capabilities of existing models and outline future directions and developments in more consistent MLLM models and pre-training objectives (*e.g.*, see efforts in language LLMs (Aggarwal et al., 2023; Chen et al., 2024; Jang & Lukasiewicz, 2023)).

Toward the latter goal, we propose a simple adapter module based strategy that effectively improves consistency. The adapter is flexible and can be added to any existing MLLM. It sits between the MLLM embedding layer and the frozen LLM decoder. The goal of the adapter is to modify the LLM's embeddings such that they are invariant to surface form variations in the language prompt / question or the image. We test the proposed adapter in widely used MLLMs such as BLIP-2 and LLaVa 1.5M models. The experiments show that it is able to improve performance on all three tasks in our proposed benchmark (shown in Figure 1 (Right)).

Our contributions are summarized as follows.

- We propose a new consistency benchmark, MM-R$^3$, that enables evaluation of MLLM's ability to produce consistent responses to a range of inputs which are identical semantically, but differ in surface form. MM-R$^3$ includes three tasks, covering visual and lingual domains.
- We conduct detailed analyses of SoTA MLLMs in accuracy and consistency on this benchmark, taking the first step towards consistency in MLLMs, which is a fundamental requirement for MLLMs to be reliably deployable for most task.
- We propose a simple but effective adapter-based strategy that can be added to any MLLM with lightweight training. Experiments on BLIP-2 and LLaVa 1.5M models show that adding the adapter significantly improves performance on our benchmark regarding consistency metrics.

## 2 RELATED WORK

**Multimodal Large Language Models (MLLMs).** The study and development of MLLMs (Liu et al., 2024; 2023a; Li et al., 2023b; Bai et al., 2023b; OpenAI, 2023; Lin et al., 2024; Chen et al., 2023; Wang et al., 2023; Sun et al., 2024; Xue et al., 2024) has recently seen a surge in popularity. Motivated by the impressive achievement made by recent LLMs (Brown et al., 2020; Chowdhery et al., 2023; OpenAI, 2023; Touvron et al., 2023; Chung et al., 2024), researchers have ventured into augmenting these models with visual capabilities to tackle multimodal tasks more effectively. A pioneering effort in this realm was the Flamingo model (Alayrac et al., 2022), which integrated the CLIP image encoder with traditional LLMs. This initiative paved the way for the development of MLLMs aimed at enhancing multimodal integration. For example, models like LLaVa 1.5M (Liu et al., 2023a; 2024), BLIP-2 (Li et al., 2023b), MiniGPT-4 (Zhu et al., 2024; Chen et al., 2023), Qwen-VL (Bai et al., 2023b), mPlug-Owl2 (Ye et al., 2024), MoE-LLaVa (Lin et al., 2024), and BLIP-3 (Xue et al., 2024), many of which leverage open-source resources to improve their ability to learn from both visual and linguistic inputs. Meanwhile, proprietary models like GPT-4V (OpenAI, 2023), Gemini (Team et al., 2023), and GPT-4o (OpenAI, 2024) have demonstrated state-of-the-art performance, pushing the boundaries of research in this field. However, as MLLMs become increasingly powerful, ensuring their reliability across both visual and linguistic domains poses a significant challenge. In this work, we aim to establish a new benchmark for evaluating the *consistency* of MLLMs, addressing this critical aspect of their development.

**Vision-Language Benchmarks.** Traditional Vision-Language (VL) benchmarks have predominantly centered on assessing performance in singular tasks, such as VQA (Antol et al., 2015; Goyal et al., 2017), OK-VQA (Marino et al., 2019), MSCOCO (Lin et al., 2014), and Visual Common-sense Reasoning (VCR) (Zellers et al., 2019). While these benchmarks are valuable, they fall short in fully gauging the comprehensive multimodal perception and reasoning capabilities of MLLMs. In response to this gap, a new wave of VL benchmarks has been developed (Fu et al., 2023; Liu et al., 2023c; Yu et al., 2023; Li et al., 2023a; Zhang et al., 2024), tailored to the complex demands of MLLMs. These benchmarks encompass a range of intricate multimodal tasks that necessitate a seamless integration of vision and language skills. For instance, MME (Fu et al., 2023) measures perceptual and cognitive skills across a total of 14 sub-tasks, offering a comprehensive measure of an MLLM's abilities. MME-RealWorld Zhang et al. (2024) is the largest manually annotated benchmark to date, focusing on real-world applications with high-resolution images. It contains 43 sub-class tasks across 5 real-world scenarios which are extremely challenging even for humans. Despite the advancements these benchmarks represent, their primary focus remains on benchmarking various skills to measure MLLMs' performance in terms of accuracy. MAD-Bench (Qian et al., 2024), on the other hand, explores MLLM robustness by examining how models reconcile discrepancies between textual prompts and images. Our work takes a different stance by prioritizing the consistency of MLLMs. In contrast to prior works, we propose a novel consistency benchmark that evaluates not just accuracy, but also the consistency of models across visual and linguistic domains.

**Consistency in Language.** Evaluating consistency in LLMs has emerged as a crucial area of inquiry, with semantic consistency being the most widely used focus in consistency analyses. This concept posits that a model should deliver consistent outcomes in semantically equivalent scenarios (Elazar et al., 2021). Notably, it has been observed that pre-trained language models may yield divergent predictions for masked language tasks when singular objects in queries are replaced with their plural counterparts (Ravichander et al., 2020) or queries are paraphrased (Elazar et al., 2021), indicating variability in response to semantically similar inputs. Building on these findings, a recent study (Jang & Lukasiewicz, 2023) comprehensively investigated LLM consistency, exploring dimensions such as logical coherence and semantic integrity, with the properties of negation, symmetry, and transitive consistency. Besides consistency evaluation, improving the robustness of the LLM has also emerged as a research challenge. Liu et al. (2023b) proposes low-parameter finetuning methods that show a better out-of-distribution performance for generation and classification tasks. Newman et al. (2022) proposes a P-Adapter that captures the factual information from the input prompts and improves the prediction consistency. Inspired by these pioneering efforts to assess LLM consistency, our approach seeks to extend this evaluation to multimodal contexts. We employ three distinct tasks, question rephrasing, image restyling, and context reasoning, to analyze the consistency of responses generated by MLLMs. This methodology not only assesses lingual semantic consistency but measures the visual comprehension consistency. Furthermore, we build an adapter to mitigate inconsistency that results from semantically equivalent lingual and visual prompts.

Table 1: **Statistics of MM-R$^3$ Benchmark.** We list the number of examples in each task as well as the source dataset from which we collect the examples. Test and Train splits are completely disjoint.

| | Task | #Examples | Source |
|---|---|---|---|
| Train | Question Rephrasing | 16, 894 | InfographicsVQA (Mathew et al., 2022), OKVQA (Marino et al., 2019) |
| | Image Restyling | 27,226 | Google Landmarks v2 (Weyand et al., 2020),Indoor Scene (Quattoni & Torralba, 2009) |
| | Context Reasoning | 30,003 | MSCOCO (Lin et al., 2014) |
| Test | Question Rephrasing | 3,516 | InfographicsVQA (Mathew et al., 2022), OKVQA (Marino et al., 2019) |
| | Image Restyling | 5,328 | Google Landmarks v2 (Weyand et al., 2020),Indoor Scene (Quattoni & Torralba, 2009) |
| | Context Reasoning | 4,500 | MSCOCO (Lin et al., 2014) |

## 3 MM-R$^3$ BENCHMARK

### 3.1 OVERVIEW OF MM-R$^3$ BENCHMARK

We introduce a new benchmark designed to assess the semantic consistency of MLLMs across both visual and linguistic domains. To achieve this, we have crafted three specific tasks: *question rephrasing*, *image restyling*, and *context reasoning*. These tasks are designed to probe the models' ability to maintain consistency in their responses. Comprehensive statistics of our benchmark are presented in Table 1 .The questions and images utilized in this benchmark are derived from existing datasets, which have been adapted to our tasks, ensuring a thorough evaluation of MLLM consistency.

### 3.2 TASKS AND DATA COLLECTION

In this section, we describe the tasks in MM-R$^3$ Benchmark and present two examples on each task.

**Question Rephrasing.** The goal of the question rephrasing task is to assess the ability of MLLMs to provide consistent responses to variously rephrased questions. For this task, we employ GPT-3.5 to generate alternate versions of given questions. The questions and images are sourced from the InfographicsVQA (Mathew et al., 2022) and OKVQA (Marino et al., 2019) datasets. To

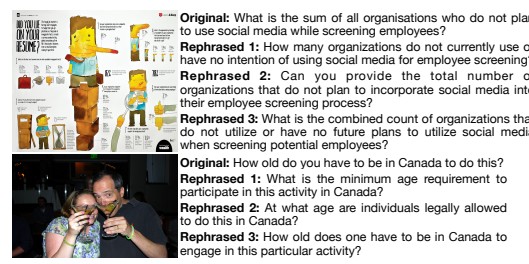

make the rephrasing meaningful, we take the questions containing more than 10 words to do the rephrasing. Using the prompt *"I have a question: ⟨Question⟩ Please give me three different types of rephrased questions to which the answer would be ⟨Answer⟩."*, we obtain three distinct rephrasings for each question. These rephrased questions, along with the corresponding images, are then presented to MLLMs to evaluate their consistency with respect to linguistic surface form perturbations. In total, we select 760 images and derive 3, 516 rephrased questions.

**Image Restyling.** Humans possess remarkable ability to recognize places and objects in images despite changes in style. We posit that MLLMs should demonstrate a similar level of adaptability. To this end, we have devised a task that assesses MLLM consistency in the face of varied image styles. To generate variations of

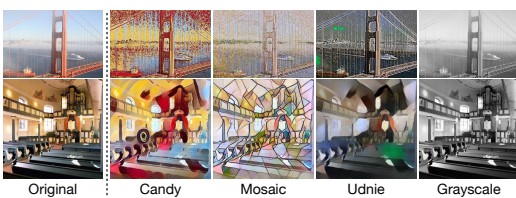

Original | Candy | Mosaic | Udnie | Grayscale

styles, we leverage the style transfer model of Johnson et al. (2016) along with Instance Normalization (Ulyanov et al., 2016). Our dataset comprises both outdoor scenes from the Google Landmarks Dataset v2 (Weyand et al., 2020) and indoor scenes from the Indoor Scene Dataset (Quattoni & Torralba, 2009). The images undergo transformations to exhibit four distinct styles: Candy, Mosaic, Udnie, and Grayscale. The MLLMs are then tasked with describing the depicted places in two sentences by using the prompt *"Please describe the place in the image in two sentences."*, based on these stylistically altered images. This approach allows us to evaluate MLLMs consistency with respect to the visual domain variations. Specifically, their ability to describe the scene in a semantically similar manner irrespective of induced image style. We collected 600 outdoor and 732 indoor scenes, resulting in a total of 5, 328 styled images.

**Context Reasoning.** When looking at an image, humans possess an intuitive ability to infer occluded elements, coming from contextual cues and commonsense reasoning about the space. This capability allows us to imagine what lies behind an obstruction, regardless of the nature of the masking element. We

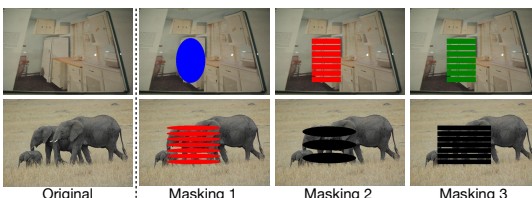

argue that MLLMs should exhibit a comparable level of perceptual inference. To evaluate this, we introduce an image masking task where objects within images are randomly masked. We select images from the MSCOCO dataset (Lin et al., 2014), ensuring that the size of the masked object relative to the entire image falls within a range of $0.1$ to $0.25$. This criterion guarantees that the masked objects are neither too small to be indiscernible nor too large to dominate the image. The masking is applied using three distinct types: lines, shapes, and colors. Specifically, we use $[1, 3, 5, 7]$ lines, rectangle and ellipse shapes, and choose from a palette of red, blue, green, yellow, white, and black colors for masking. Then, we present the masked images to MLLMs alongside the query: *"What kind of object is in the masked region?"* This setup enables us to measure the MLLMs' consistency in reasoning and handling masking variations, thereby testing their inference capabilities. In the end, $1,500$ images were selected, resulting in a total of $4,500$ uniquely masked images.

**Semantic equivalence to original data.** We conduct human evaluations to quantify the quality of rephrased questions and restyled images. We do so using forced choice experiment on $100$ randomly sampled question-rephrasing pairs, and $100$ stylized images. We find $92\%$ language rephrasing and $86\%$ image restyling have semantic equivalence for humans; validating the quality of our dataset.

### 3.3 EVALUATION METHOD

The responses expected from MLLMs in our tasks are inherently open-ended, reflecting the diverse capabilities of these models. However, the design of our questions is meticulously aligned with the ground truth in the original dataset annotations. This alignment enables us to leverage the annotations effectively during evaluation. To systematically assess the performance of MLLMs, we introduce four distinct evaluation methods, each capturing different aspects of model performance.

**Accuracy (Acc).** The evaluation of accuracy is conducted through a straightforward method: we assess the responses from MLLMs based on an exact match criterion with the ground truth annotations. Specifically, if the ground truth annotation is encompassed within the MLLM's response, we consider the response correct. The accuracy score is then calculated as the average of correct responses across the benchmark dataset, providing a measure of the MLLMs' performance.

**Similarity with GT ($S_{GT}$).** Given the limitations of the exact match criterion, which may inaccurately categorize semantically similar responses as incorrect, we introduce a similarity metric designed to evaluate the nuanced semantic parallels between MLLMs' responses and the ground truth annotations. For instance, terms like *person* share semantic similarity with *man* and *woman*, yet would be deemed incorrect under a strict accuracy metric. To address this, our approach utilizes semantic similarity metric (Reimers & Gurevych, 2019), which leverages LLM encodings to assess the similarity between responses and target answers. This metric calculates the similarity score of an MLLM response and its corresponding annotation, with the overall performance represented by the average of these scores across the dataset. This metric provides a more subtle assessment of model, rewarding semantic accuracy over mere lexical matching.

**Consistency Accuracy (Con).** This metric is designed to quantify the proportion of responses that manifest a specified degree of semantic consistency. To achieve this, we leverage semantic similarity metrics, Sentence-Similarity of Reimers & Gurevych (2019), to compute the pairwise similarity scores between responses, utilizing a threshold of $0.7$ to delineate semantic consistency. The threshold is based on the observation of Semantic Textual Similarity benchmark (Cer et al., 2017). A response is deemed consistent if its similarity score with a paired response surpasses this threshold. The metric's final value is obtained by averaging the proportion of consistent responses across the entire dataset, providing an aggregate measure of semantic consistency within the MLLMs' outputs.

**Consistency Similarity ($S_C$).** Similar to the Consistency Accuracy metric, we measure the consistency similarity by calculating pairwise similarity scores between responses. Instead of setting a threshold, we derive the final metric by computing the average of these similarity scores across the entire dataset. This allows us to more *continuously* asses the coherence of responses.

Table 2: Overall results of MLLMs on the **Question Rephrasing Task**. The numbers in red indicate the difference between *Sampling* and *All* results. The best-performing model in each metric is in bold and the second-best model is underlined.

| | Models | Sampling | | | | All | | | |
|---|---|---|---|---|---|---|---|---|---|
| | | Acc | $S_{GT}$ | Con | $S_C$ | Acc | $S_{GT}$ | Con | $S_C$ |
| Open-sourced | BLIP-2 | 30.29 | 56.25 | 100.00 | 100.00 | 25.13 (-5.16) | 52.91 (-3.34) | 48.15 (-51.85) | 63.90 (-36.10) |
| | mPLUG-Owl2 | 40.10 | 70.12 | 58.58 | 73.26 | 30.44 (-9.67) | 61.10 (-9.03) | 36.71 (-21.86) | 55.63 (-17.63) |
| | LLaVa 1.5M | 40.02 | 70.91 | 100.00 | 100.00 | 31.01 (-9.01) | 62.85 (-8.06) | 48.47 (-51.53) | 63.99 (-36.01) |
| | MoE-LLaVa | 34.47 | 65.94 | 81.83 | 87.48 | 28.85 (-5.62) | 57.97 (-7.97) | 45.32 (-36.51) | 61.16 (-26.32) |
| | Qwen-VL-Chat | 52.05 | 82.08 | 93.25 | 95.20 | 36.31 (-15.73) | **70.21** (-11.87) | 55.34 (-37.91) | 69.70 (-25.51) |
| | BLIP-3 | 32.70 | 55.44 | 80.00 | 80.00 | 30.94 (-1.76) | 60.98 (5.54) | 47.70 (-32.3) | 63.99 (-16.01) |
| Closed- | Gemini | 47.50 | 71.98 | 86.75 | 90.48 | 39.35 (-8.15) | 66.22 (-5.76) | 58.26 (-28.49) | 70.66 (-19.82) |
| | GPT-4V | 53.26 | 64.75 | 76.09 | 81.58 | **50.22** (-3.04) | 67.27 (2.52) | 55.26 (-20.83) | 69.18 (-12.41) |
| | GPT-4o | 57.06 | 76.53 | 85.42 | 89.06 | 46.99 (-10.07) | 69.04 (-7.49) | **60.87** (-24.55) | **72.01** (-17.05) |

## 4 EVALUATION RESULTS

### 4.1 MODELS

All experiments are conducted on NVIDIA A40 GPUs. We evaluate a total of six widely used, open-sourced MLLMs, including BLIP-2 (Li et al., 2023b), mPLUG-Owl2 (Ye et al., 2024), LLaVa 1.5M (Liu et al., 2024; 2023a), MoE-LLaVa (Lin et al., 2024), Qwen-VL-Chat (Bai et al., 2023b), and BLIP-3 (Xue et al., 2024) on our consistency benchmark. Additionally, the proprietary models, Gemini (Team et al., 2023), GPT-4V (OpenAI, 2023), and GPT-4o (OpenAI, 2024), are included to enrich our comparative analysis. Details of these models are provided in the Appendix A.1.

### 4.2 MAIN RESULTS

The main results for different tasks are shown in Tables 2, 3, and 4. For each task, we present results under Sampling and All Data. *Sampling* denotes querying the MLLM model multiple times with the identical input image and question to observe natural variation in responses without changing the conditions. This allows us to measure how much difference comes from the stochasticity of the model versus the rephrasing of the condition. For each example, we query the MLLM model four times and average across the whole dataset. *All* represents the mean performance across the dataset. Additional analysis in terms of consistency for correct vs. incorrect answers is given in Appx A.5. These settings allow us to examine the relationship between model consistency and accuracy.

Our key findings present intriguing insights. Notably, we observe a divergence between accuracy and consistency across the three tasks. While accuracy performance remains relatively similar and competitive among the different models, there is a marked disparity in their consistency performances. Moreover, we note significant variability in consistency across models, in contrast to the more stable accuracy performance. Furthermore, the variations in consistency are significantly more substantial in the image restyling and context reasoning tasks, suggesting that changes within the visual domain have a greater influence on consistency than linguistic modifications.

### 4.3 DETAILED ANALYSIS

#### 4.3.1 ANALYSIS ON EACH TASK

- **Question Rephrasing:** We provide quantitative results in Table 2. In the question rephrasing task, GPT-4V achieves the highest accuracy among the nine evaluated models, with Qwen-VL-Chat outperforming the rest of the open-source models. BLIP-2 and LLaVa 1.5M, set with a temperature of 0, achieve perfect consistency in sampling at $100\%$. However, their performance significantly drops when faced with rephrased questions, indicating a lower consistency in adapting to question variations compared to other models. It implies that BLIP-2 and LLaVa 1.5M might be sensitive to the input prompts. Another interesting observation is that, although Qwen-VL-Chat ranks lower in accuracy compared to GPT-4V, it is better in consistency metrics.

  A qualitative example is illustrated in Figure 5. Although the Qwen-VL model does not answer correctly, it consistently generates answers "columbia". In contrast, LLaVa 1.5M and BLIP-2 show variability in their responses, highlighting lower consistency and sensitivity to prompting.

Table 3: Overall results of MLLMs on the **Image Restyling Task**. We show the results on the entire dataset and Sampling variations with the difference highlighted in red/green color. The best-performing model in each metric is in bold and the second-best model is underlined.

| | Models | \multicolumn{4}{c}{Sampling} | | | | \multicolumn{4}{c}{All} | | | |
|---|---|---|---|---|---|---|---|---|---|
| | | Acc | $S_{GT}$ | Con | $S_C$ | Acc | $S_{GT}$ | Con | $S_C$ |
| Open-Sourced | BLIP-2 | 16.82 | 16.40 | 100.00 | 100.00 | 13.01 (-3.81) | 17.02 (0.62) | 38.36 (-61.64) | 62.83 (-37.17) |
| | mPLUG-Owl2 | 15.71 | 14.12 | 53.47 | 69.73 | 8.95 (-6.76) | 15.25 (1.13) | 26.29 (-27.18) | 59.21 (-10.52) |
| | LLaVa 1.5M | 15.24 | 15.07 | 100.00 | 100.00 | 10.47 (-4.77) | 15.49 (0.42) | 50.08 (-49.92) | 68.46 (-31.54) |
| | MoE-LLaVa | 15.99 | 15.48 | 95.75 | 86.36 | 12.93 (-3.06) | 16.60 (1.12) | 70.27 (-25.48) | 74.79 (-11.57) |
| | Qwen-VL-Chat | 13.81 | 15.42 | 77.28 | 82.53 | 8.28 (-5.54) | 15.73 (0.31) | 23.10 (-54.18) | 53.29 (-29.24) |
| | BLIP-3 | 17.12 | 15.08 | 100.00 | 100.00 | 11.92 (-5.20) | **17.31** (2.23) | 51.39 (-48.61) | 68.20 (-31.80) |
| Closed- | Gemini | 14.47 | 16.55 | 75.85 | 75.88 | 13.68 (-0.79) | 16.15 (-0.4) | 53.37 (-22.48) | 68.95 (-6.93) |
| | GPT-4V | 25.94 | 15.34 | 97.71 | 85.07 | 9.72 (-16.22) | 15.90 (0.56) | 52.55 (-45.16) | 66.25 (-18.81) |
| | GPT-4o | 16.03 | 17.65 | 96.65 | 84.20 | **16.31** (0.28) | 16.83 (-0.82) | **79.60** (-17.05) | **77.07** (-7.13) |

- **Image Restyling:** The results are shown in Table 3. In the image restyling task, GPT-4o stands out across three metrics and MoE-LLaVa leads among open-sourced models. Although BLIP-2 outperforms other models in accuracy, its consistency is exceptionally poor.

  Figure 5 presents a comparative example of responses from various MLLMs. The BLIP-2 model typically produces brief yet accurate answers. Conversely, BLIP-3 model offers more detailed descriptions, enhancing the comprehensiveness of the place's understanding. Although these detailed descriptions result in a lower similarity score with ground truth (lower performance on $S_{GT}$), they demonstrate a higher consistency across different responses.

- **Context Reasoning:** Table 4 presents the performance of various MLLMs in the image masking task. Among the evaluated MLLMs, the BLIP-2 and LLaVa 1.5M models achieve a better performance in consistency metrics. On the other hand, the Qwen-VL-Chat and BLIP-3 model show the weakest performance across the MLLMs, with a notable drop in consistency metrics, which possibly means a lesser capability for abductive reasoning compared to its counterparts.

  Figure 5 shows an example of the Context Reasoning task. Gemini and GPT-4o generate a semantically similar response as ground truth and provide detailed rationales in their responses, highlighting the strength of abductive reasoning. An interesting observation is the models' tendency to predict line-shaped masks as bats, which suggests a potential model bias.

### 4.3.2 ANALYSIS ACROSS TASKS.

Across three tasks, the Qwen-VL-Chat model performs better in the Question Rephrasing tasks but falls short in the Image Restyling and Context Reasoning tasks among the open-sourced models. This disparity may come from its use of the state-of-the-art LLM, Qwen (Bai et al., 2023a), which likely provides Qwen-VL-Chat with superior initial language representations. On the other hand, BLIP-2 achieve the best performance in the Context Reasoning. This is possibly due to its unique image-text learning during the pre-training phase, which could facilitate a better contextual understanding of images. We believe the matching between image and language indeed helps the models learn the context in the image. Meanwhile, LLaVa 1.5M and MoE-LLaVa exhibit balanced performances across all tasks, achieving a good trade-off between accuracy and consistency. Among the closed-source models, GPT-4o outperforms Gemini and GPT-4V in all three tasks on accuracy and consistency. However, a notable observation is that current top-performing MLLMs still drop significantly in terms of consistency when facing changes in visual and linguistic domains. The gaps suggest that further effort is needed to enhance the performance of MLLMs in terms of consistency.

### 4.3.3 ANALYSIS ON DIFFERENT RESOLUTIONS IN IMAGE RESTYLING TASK

In the Image Restyling task, we notice that different resolutions of original images might yield different levels of detail when styling the images. In the Image Restyling task figure, the Golden Gate has a higher resolution, so it shows more details after restyling. Conversely, the bottom row—the church—has a lower resolution and shows fewer details. To analyze the effect of detail levels for MLLMs, we resize the images before performing the style transfer. We resize the images to low $(224 \times 224)$, mid $(640 \times 640)$, and high $(1024 \times 1024)$ resolutions. The results are shown in Figure 2. Compared with the original Image Styling task results (Table 3), the low-resolution results drop significantly, especially in the **Con** and $S_C$ metrics, indicating that the level of detail affects consistency the most. On the other hand, the mid and high-resolution results show consistent improvements, indicating that the consistency of models increases with increase in resolution.

Table 4: Overall results of model performances on the **Context Reasoning Task**, with red/green numbers showing differences between *Sampling* and *All* results. The top model for each metric is highlighted in bold and the second-tier model is underlined.

| | Models | Sampling | | | | All | | | |
|---|---|---|---|---|---|---|---|---|---|
| | | Acc | $S_{GT}$ | Con | $S_C$ | Acc | $S_{GT}$ | Con | $S_C$ |
| Open-Sourced | BLIP-2 | 28.20 | 39.10 | 100.00 | 100.00 | 27.91 (-0.29) | 38.97 (-0.13) | **82.44** (-17.56) | **88.80** (-11.20) |
| | mPLUG-Owl2 | 24.43 | 35.21 | 33.36 | 53.25 | 24.47 (0.03) | 35.34 (0.12) | 27.64 (-5.71) | 47.97 (-5.29) |
| | LLaVa 1.5M | 28.34 | 42.54 | 100.00 | 100.00 | 28.67 (0.32) | 42.52 (-0.03) | 68.04 (-31.96) | 77.02 (-22.98) |
| | MoE-LLaVa | 26.13 | 38.91 | 75.49 | 81.08 | 25.16 (-0.98) | 38.47 (-0.45) | 39.40 (-36.09) | 56.86 (-24.22) |
| | Qwen-VL-Chat | 20.03 | 32.30 | 75.72 | 83.19 | 20.11 (0.08) | 32.20 (-0.10) | 30.69 (-45.03) | 48.95 (-34.24) |
| | BLIP-3 | 28.00 | 36.50 | 100.00 | 100.00 | 27.96 (-0.04) | 36.48 (-0.02) | 40.02 (-50.98) | 54.88 (-45.12) |
| Closed- | Gemini | 55.60 | 57.32 | 68.08 | 78.99 | **55.31** (-0.29) | **56.35** (-0.97) | 45.22 (-22.86) | 62.66 (-16.33) |
| | GPT-4V | 33.72 | 20.97 | 37.21 | 58.19 | 32.53 (-1.19) | 21.25 (0.28) | 34.37 (-2.84) | 57.05 (-1.14) |
| | GPT-4o | 52.10 | 31.82 | 59.71 | 70.50 | 51.73 (-0.37) | 31.67 (-0.15) | 49.49 (-10.22) | 66.82 (-3.68) |

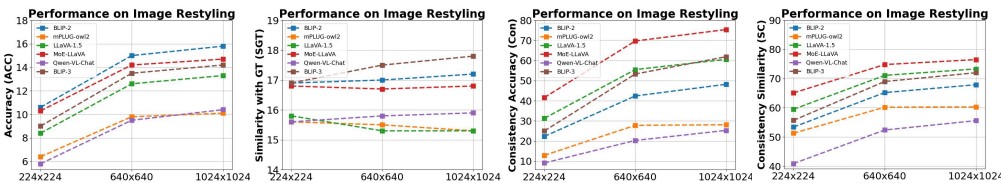

Figure 2: Stylization with different resolutions on the **Image Restyling Task**.

Table 5: **Impact of Model Size.** Different MLLM model sizes on MM-R$^3$ benchmark. The $^*$ denotes that we ignore generated responses that have an empty output.

| | Question Rephrasing | | | | Image Restyling | | | | Context Reasoning | | | |
|---|---|---|---|---|---|---|---|---|---|---|---|---|
| Models | Acc | $S_{GT}$ | Con | $S_C$ | Acc | $S_{GT}$ | Con | $S_C$ | Acc | $S_{GT}$ | Con | $S_{Con}$ |
| BLIP-2 (opt2.7B) | 19.0$^*$ | 47.3$^*$ | 39.1$^*$ | 54.3$^*$ | 11.6$^*$ | 55.5$^*$ | 48.5$^*$ | 58.0$^*$ | 27.8 | 39.0 | 76.2 | 86.2 |
| BLIP-2 (flant5xxl) | 25.1 | 52.9 | 48.2 | 63.9 | 13.0 | 17.0 | 38.4 | 62.8 | 27.9 | 39.0 | 82.4 | 88.8 |
| LLaVa 1.5M (7B) | 31.0 | 62.9 | 48.5 | 64.0 | 10.5 | 15.5 | 50.1 | 68.5 | 28.7 | 42.5 | 68.0 | 77.0 |
| LLaVa 1.5M (13B) | 33.0 | 63.7 | 49.5 | 64.5 | 10.6 | 16.1 | 67.4 | 75.5 | 34.9 | 45.0 | 64.6 | 74.5 |

### 4.3.4 ANALYSIS ON MODEL SIZE

Previous studies show that the number of parameters in MLLMs affects performance in downstream tasks. For example, BLIP-2, the model achieves greater performance when the number of parameters is larger in VQA, Image Captioning, and Image-Text Retrieval Li et al. (2023b). A similar trend is found in the LLaVA 1.5M model Liu et al. (2024). As a result, we are curious how consistency is impacted with different sizes of models. We evaluate the BLIP-2 and LLaVA 1.5M models with small and large numbers of parameters. As shown in Table 5, large models outperform small models on **Acc** and $S_{GT}$ in all three tasks. However, consistency metrics (**Con** and $S_C$) do not show a similar trend. In the Question Rephrasing and the Image Restyling task, the large model performs better, while the trend is not the same in the Context Reasoning task. Hence, we observe that unlike accuracy, consistency does not always improve with increase in model size.

### 4.3.5 ANALYSIS ON DIFFERENT ENTROPY PARAMETERS

The temperature parameter in MLLMs controls the level of randomness in the model's output. Lower temperature yields more deterministic outputs, and higher temperature shows more diversity outputs. Hence, we analyze how temperature affects the consistency output in three tasks. We show the results on 3 different temperatures in Figure 3. We set the temperatures to 0.2, 0.7, and 1 on all open-sourced models. We notice that mPLUG-Owl2 and MoE-LLaVa model performances drop significantly when the temperature increases while Qwen-VL-Chat show more consistent results.

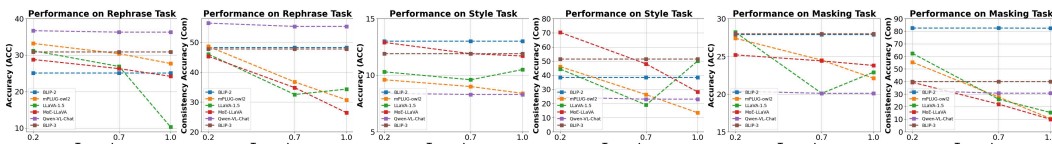

Figure 3: **Impact of Entropy.** Different entropy parameters on all three tasks for models tested.

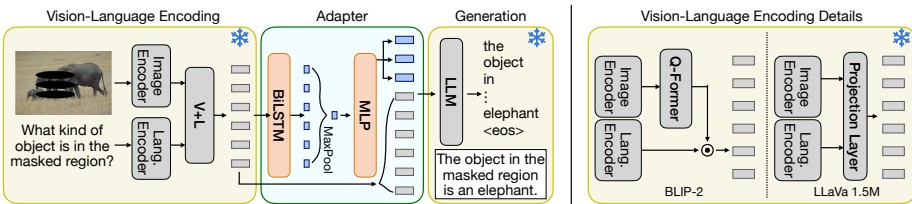

Figure 4: (Left) **Model Overview.** The adapter is added between the Vision-Language Encoding and Generation models. The encoding and decoding parts are frozen during the training of the adapter. (Right) Vision-Language Encoding details of BLIP-2 and LLaVa 1.5M.

Table 6: **Improvement Results on Three Tasks.** The consistency metrics on three tasks on both BLIP-2 and LLaVa 1.5M models significantly improved after adding the adapter.

| | Models | \multicolumn{4}{c}{Question Rephrasing} | | | | \multicolumn{4}{c}{Image Restyling} | | | | \multicolumn{4}{c}{Context Reasoning} | | | |
| | | Acc | $S_{GT}$ | Con | $S_C$ | Acc | $S_{GT}$ | Con | $S_C$ | Acc | $S_{GT}$ | Con | $S_C$ |
|---|---|---|---|---|---|---|---|---|---|---|---|---|---|
| Ori. | BLIP-2 | 25.1 | 52.9 | 48.2 | 63.9 | 13.0 | 17.0 | 38.4 | 62.8 | 27.9 | 39.0 | 82.4 | 88.8 |
| | LLaVa 1.5M | 26.9 | 59.2 | 32.5 | 53.8 | 9.6 | 14.9 | 19.0 | 56.9 | 20.1 | 28.9 | 25.9 | 42.3 |
| Adapt | BLIP-2 | 27.6 | 64.7 | 61.8 | 73.7 | 36.7 | 25.6 | 49.0 | 64.1 | 54.6 | 52.4 | 90.1 | 94.7 |
| | LLaVa 1.5M | 31.4 | 65.9 | 43.2 | 62.3 | 18.1 | 28.1 | 32.6 | 52.6 | 58.6 | 73.8 | 62.0 | 75.5 |

## 5 CONSISTENCY IMPROVEMENT

The analyses on the MM-$R^3$ benchmark in Section 4 show that consistency drops significantly with semantically equivalent prompts. To overcome this issue, we propose a simple strategy based on an adapter module that can be plugged into any MLLM with lightweight training (see Figure 4 (green)). The goal of the adapter is to help MLLMs overcome the variability of visual and language prompts by making them invariant to surface form variability in data and ensuring a single consistent output.

### 5.1 APPROACH

The design of the adapter is inspired by Newman et al. (2022). The adapter model takes the embeddings of the input prompts and outputs a new sequence of continuous embeddings that are used as input to the LLM decoder for generation. Specifically, the adapter takes the outputs of the Vision-Language encoder, passes them through a Bi-LSTM module and a max-pooling procedure to obtain the global embeddings from the input tokens. Then, an MLP is added afterward to project the max-pooling output to the size of the prefix that will be added in front of the original embeddings. In this way, the adapter not only captures the semantic embeddings from the prompts but also retains all of the original embeddings for the LLM decoder. During fine-tuning, we update only the parameters of the adapter while freezing the vision-language encoder and the language decoder. As shown in Figure 4, the yellow parts are frozen, and the green one represents the training component.

### 5.2 EXPERIMENTS

**Training Data.** We leverage the same data generation pipeline as the MM-$R^3$ benchmark to generate training data samples. Note that these training samples are completely disjoint with the samples in MM-$R^3$ benchmark which we evaluate on. In total, there are $16,894$ rephrased question-image pairs for the Question Rephrasing Task, $27,226$ styled images for the Image Restyling Task, and $30,003$ masked images for the Context Reasoning Task for training.

**Implementation Details.** We select BLIP-2 and LLaVa 1.5M for consistency improvement experiments since they are widely used in recent works, have low consistency compared to other models across lingual changes and allow us to show efficacy of our approach on different types of MLLMs families (i.e ones that use only CLIP vs Qformer based architectures). For both BLIP-2 and LLaVa 1.5M experiments, the Bi-LSTM includes 2 recurrent layers with the hidden size is set to $4096$ and dropout rate $0$. We follow the same setting as in Newman et al. (2022) to set the prefix size to 3. The model is optimized by *CrossEntropyLoss*. The initial linear rate is set to $1e-5$. A linear decay learning rate schedule is also used. We conduct all experiments on a Nvidia A40 GPU with batch size 2 on both models. The adapter is added on the top of BLIP-2 `blip2-t5 pretrain-flant5xxl` and LLaVa 1.5M `llava-v1.5-7b` models.

**Results.** The performance with and without our adapter, on all three tasks, is illustrated in Table 6. In *question rephrasing* the accuracy of the model with the adapter is marginally better (improvement

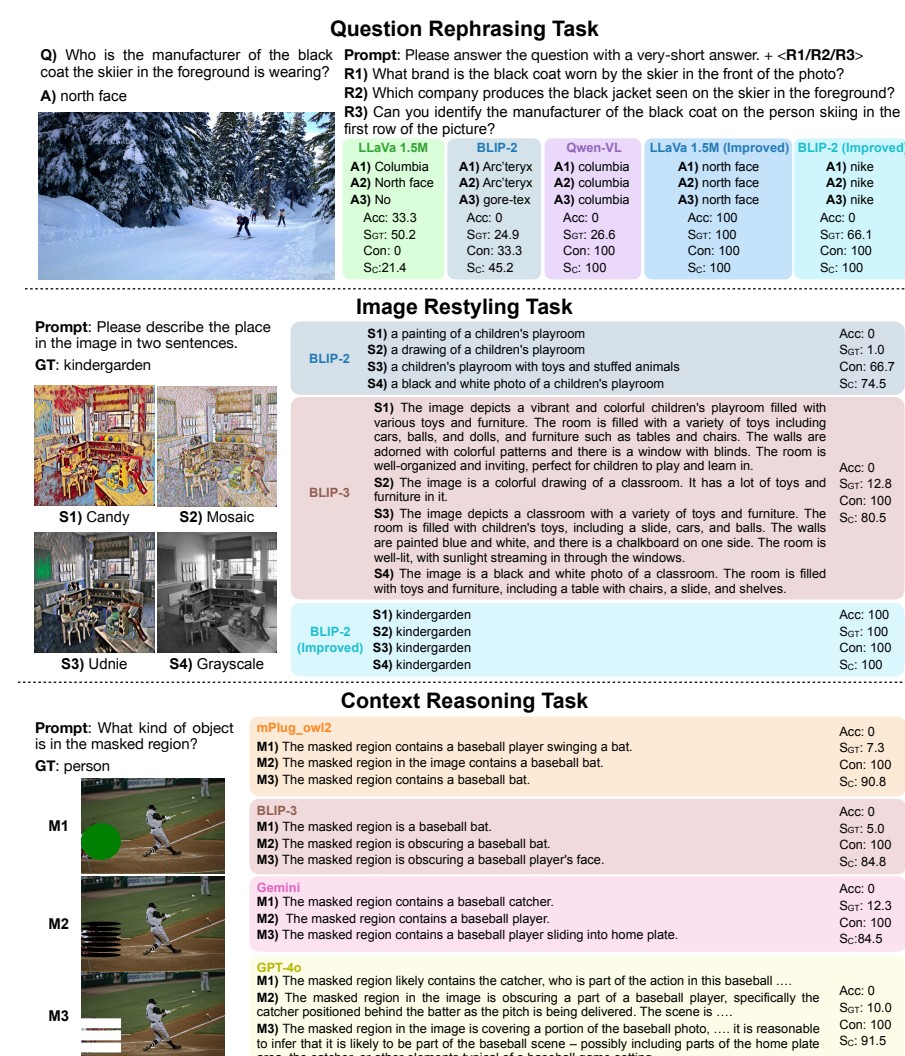

Figure 5: **Qualitative Results.** Metrics shown are computed for specific qualitative examples shown in the figure. See Appendix for more.

of +2.5 for BLIP-2 and +4.5 for LLaVa 1.5M), however, the consistency shows a very significant margin of improvement (+13.6 and +10.8 respectively). This is consistent with our earlier observation that accuracy and consistency are not necessarily aligned. The original MLLMs are already well trained for VQA tasks and so large accuracy boost is not expected. However, they are lacking in consistency (see Section 4), hence the large improvement on those metrics. For *image restyling* and *context reasoning* we do see a large improvement on both the accuracy and consistency. This is largely because original MLLMs are not trained on data of this form. Hence, the introduced adapter can both fine-tune performance on the new data *and* improve consistency on them at the same time.

## 6 CONCLUSION

In this paper, we explore and analyze *consistency* of MLLM models and its relationship to *accuracy*. We do so by introducing the MM-R$^3$ Benchmark, comprising three tasks – Question Rephrasing, Image Restyling, and Context Reasoning – to conduct a comprehensive analysis of SoTA MLLMs in terms of both accuracy and consistency. Our analysis reveals that higher accuracy does not necessarily equate to greater consistency in models, and vice versa. In addition, we observe significant variations in the consistency of SoTA models, while their accuracy levels tend to be more stable across models. These findings suggest that future MLLM development and objectives might benefit from a dual focus, emphasizing the optimization of consistency and the improvement of overall accuracy. Furthermore, we propose an effective adapter to improve consistency performance. The experiments on BLIP-2 and LLaVa 1.5M models illustrate the improved performance on three tasks.

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

# A  APPENDIX

In the appendix, we present additional details and results to complement the main paper.

## A.1  DETAILS OF MLLMs

**BLIP-2** (Li et al., 2023b): The BLIP-2 model addresses the modality gap through a novel lightweight Querying Transformer, the Q-former, leveraging a two-stage pre-training approach. Despite its lack of multimodal instruction tuning, BLIP-2 retains the LLM's capacity for following instructions. For our evaluations, we employed the `BLIP-2 FlanT5XXL` in our evaluations.

**mPLUG-Owl2** (Ye et al., 2024): mPLUG-Owl2 utilizes a modularized network design tailored for handling multi-modal inputs. It contains a modality-adaptive module to align different modalities into a shared semantic space for relational learning. The model's architecture, including the visual encoder, visual abstractor, and language decoder, are all updated during training. We use `mplug-owl2-llama2-7b` for inference and our performance analysis.

**LLaVa 1.5M** (Liu et al., 2024; 2023a): LLaVa is an end-to-end model combining LLaMA/ Vicuna on GPT-generated multimodal instruction-following data. It provides general-purpose visual and language understandings, boasting chat capabilities that parallel the multimodal proficiency of GPT-4. We conducted our experiments using the `llava-v1.5-7b` version.

**MoE-LLaVa** (Lin et al., 2024): The MoE-LLaVa model incorporates a Mixture of Experts (MoE) architecture with learnable routers, comprising multiple sparse paths that uniquely activate only the top-k experts during deployment while keeping the remaining experts inactive. This design allows MoE-LLaVa to achieve performance comparable to other high-end MLLMs with the provided sparse path. We use `MoE-LLaVA-Phi2-2.7B-4e-384` version for evaluation.

**Qwen-VL-Chat** (Bai et al., 2023b): Qwen-VL-Chat builds upon the foundation of Qwen-VL, with training that encompasses not just traditional image descriptions and question-answering tasks, but also extends to grounding and text-reading capabilities through the alignment of image-caption-box tuples. The version tested and analized in our experiments is `Qwen-VL-Chat`.

**BLIP-3** (Xue et al., 2024): BLIP-3 (xGen-MM) consists of a Vision Transformer, a vision token sampler to downsample the image embeddings, and a pre-trained Large Language Model. BLIP-3 enables higher-resolution images as inputs by using patch-wise encoding. The patch-wise encoding preserves the resolution of the original images by encoding the split image patches separately. Then a perceiver resampler is used to downsample the visual tokens before sending them to the LLM. This design allows BLIP-3 to reduce the sequence length of vision tokens while keeping the higher-resolution images as inputs. We evaluate the Consistency Benchmark on `xgen-mm-phi3-mini-instruct-r-v1` version.

**Gemini** Team et al. (2023): Gemini is developed by Google. It is a multimodal model using a Transformer to process various inputs, such as text, images, audio, and video. Unlike models using separate modules to encode different types of data, Gemini uses a shared Transformer. This enables the model to leverage shared reasoning capabilities. We use `gemini-1.5-flash` for evaluation.

**GPT-4V** (OpenAI, 2023): GPT-4V extends the capabilities of Generative Pre-trained Transformers by integrating visual understanding, enabling it to process and generate content based on both textual and visual inputs. The development represents a significant advancement in AI, making it a versatile tool for a wide range of applications that require the understanding of both text and imagery. We evaluate the Consistency Benchmark on `gpt-4-vision-preview` version.

**GPT-4o** (OpenAI, 2024): GPT-4o ("o" for "omni") is the most advanced model released by OpenAI. It accepts multimodal inputs, e.g. texts and images. We leverage `gpt-4o` version to evaluate the Consistency Benchmark.

## A.2  SIMILARITY METRICS DETAILS

For evaluating Similarity with Ground Truth ($S_{GT}$), Consistency Accuracy (Con), and Consistency Similarity ($S_C$), we leverage semantic similarity metrics, Sentence-Similarity Reimers & Gurevych (2019). This metric utilizes large language model encodings to compare the semantic content of

predicted and target texts. Specifically, Sentence-Similarity achieves this by transforming input texts into embeddings via a pre-trained Transformer model, capturing their semantic differences. The similarity between these embeddings, and thus the texts, is calculated using cosine similarity, providing a measure of their semantic closeness.

### A.3 POST-PROCESSING OF ANSWERS

In the Context Reasoning task, we notice that MLLMs often repeat phrases from the question, such as "the masked region is..." which raises the consistency scores due to these repetitive terms when calculating answer similarity. However, our primary interest lies in identifying the objects behind the masks. To address this, we conduct post-processing on the responses before metric calculation. This involves filtering out common words like *there*, *is*, *of*, *in*, *the*, etc., and phrases frequently repeated from the question, such as *masked region* and *image*. This adjustment allows for a more fair comparison by focusing on the substance of the responses rather than their structural similarities.

### A.4 DATA GENERATIONS AND TASKS PIPELINE

We provide a comprehensive pipeline that details both the data collection process and the methods of prompting MLLMs for each task, shown in Figure 6.

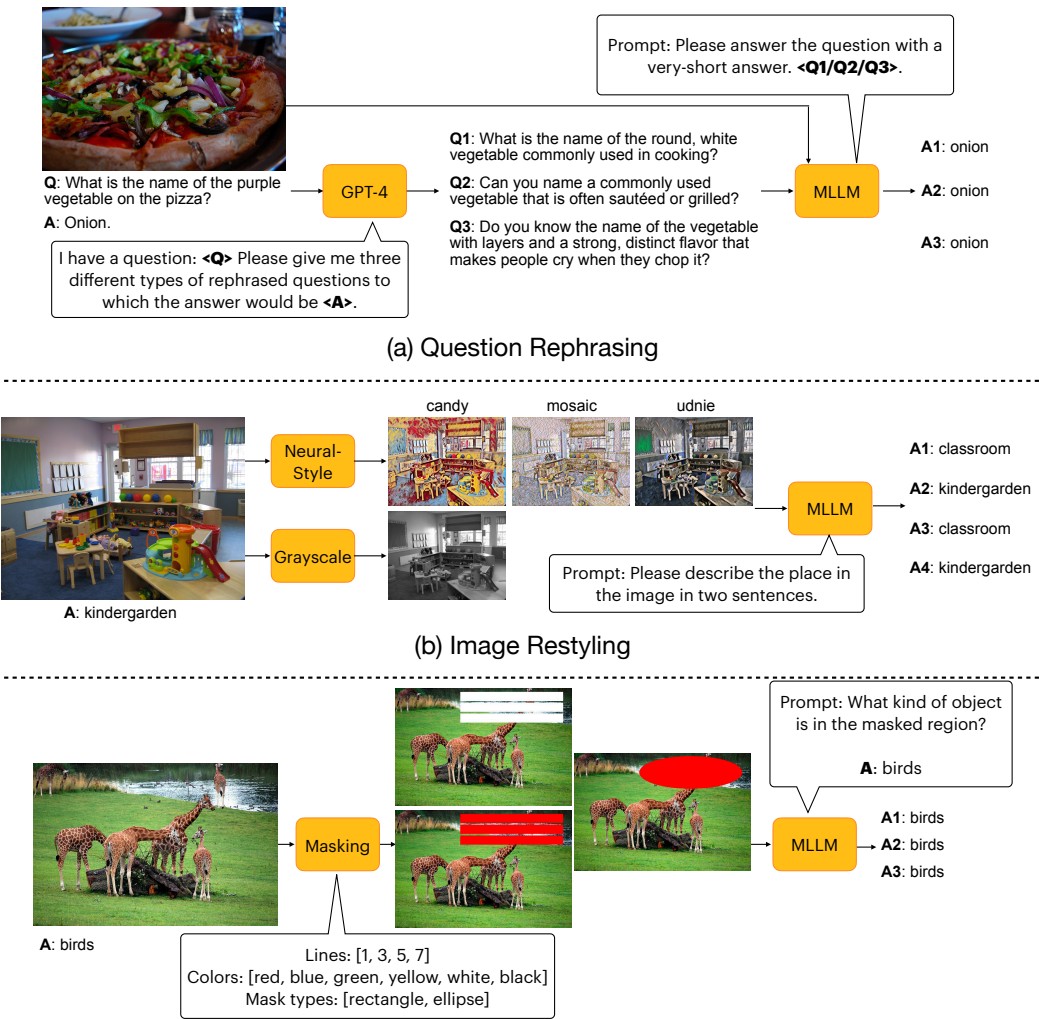

Figure 6: The pipeline for the data collection process and prompting MLLMs for each task.

## A.5 FAILURE CASES

*Failure Cases* means we only focus on the answers that are not fully correct under different changes.

Table 7: Failure results of different models.

| | Models | Question Rephrasing Task | | | Image Restyling Task | | | Context Reasoning Task | | |
|---|---|---|---|---|---|---|---|---|---|---|
| | | $S_{GT}$ | Con | $S_C$ | $S_{GT}$ | Con | $S_C$ | $S_{GT}$ | Con | $S_C$ |
| Open-Sourced | BLIP-2 | 46.44 | 41.75 | 59.33 | 16.87 | 37.15 | 62.01 | 28.68 | 77.78 | 86.41 |
| | mPLUG-Owl2 | 54.58 | 27.96 | 49.25 | 15.23 | 25.46 | 58.90 | 31.05 | 23.29 | 44.81 |
| | LLaVa 1.5M | 56.26 | 39.83 | 57.99 | 15.43 | 48.72 | 67.86 | 33.28 | 62.34 | 73.14 |
| | MoE-LLaVa | 52.35 | 38.67 | 56.18 | 16.61 | 69.09 | 74.44 | 33.06 | 33.38 | 53.01 |
| | Qwen-VL-Chat | 62.17 | 44.15 | 62.13 | 15.73 | 22.48 | 52.91 | 28.23 | 26.17 | 45.70 |
| | BLIP3 | 55.56 | 41.23 | 59.49 | 17.29 | 50.64 | 67.85 | 31.22 | 34.50 | 50.61 |
| Closed- | Gemini | 58.22 | 47.47 | 63.32 | 16.13 | 51.89 | 68.52 | 45.24 | 28.63 | 51.14 |
| | GPT-4V | 52.43 | 34.46 | 54.67 | 15.90 | 51.43 | 65.83 | 17.22 | 29.44 | 54.64 |
| | GPT-4o | 59.85 | 47.01 | 62.76 | 16.78 | 78.34 | 76.67 | 25.01 | 41.72 | 64.27 |

## A.6 DETAILS OF DIFFERENT RESOLUTIONS IN IMAGE RESTYLING TASK

The numbers in Table 8 correspond to Figure 2.

Table 8: Stylization with different resolutions on the **Image Restyling Task**.

| Models | 224× 224 | | | | 640× 640 | | | | 1024× 1024 | | | |
|---|---|---|---|---|---|---|---|---|---|---|---|---|
| | Acc | $S_{GT}$ | Con | $S_C$ | Acc | $S_{GT}$ | Con | $S_C$ | Acc | $S_{GT}$ | Con | $S_{Con}$ |
| BLIP-2 | 10.6 | 16.9 | 22.3 | 53.4 | 15.0 | 17.0 | 42.4 | 65.2 | 15.8 | 17.2 | 48.2 | 67.9 |
| mPLUG-Owl2 | 6.4 | 15.6 | 12.9 | 51.3 | 9.8 | 15.5 | 27.8 | 60.2 | 10.1 | 15.3 | 28.1 | 60.3 |
| LLaVa 1.5M | 8.4 | 15.8 | 31.3 | 59.5 | 12.6 | 15.3 | 55.5 | 71.1 | 13.3 | 15.3 | 60.6 | 73.3 |
| MoE-LLaVa | 10.3 | 16.8 | 41.6 | 65.1 | 14.2 | 16.7 | 69.7 | 74.8 | 14.7 | 16.8 | 75.4 | 76.5 |
| Qwen-VL-Chat | 5.8 | 15.6 | 9.1 | 40.9 | 9.5 | 15.8 | 20.3 | 52.4 | 10.4 | 15.9 | 25.3 | 55.6 |
| BLIP-3 | 9.0 | 16.9 | 25.0 | 55.7 | 13.5 | 17.5 | 53.3 | 69.0 | 14.2 | 17.8 | 61.9 | 72.0 |

## A.7 DETAILS OF DIFFERENT ENTROPY PARAMETERS

We show the actual numbers in Figure 3 in Table 9.

Table 9: Different entropy parameters on three tasks.

| Question Rephrasing Task | | | | | | | | | | | | |
|---|---|---|---|---|---|---|---|---|---|---|---|---|
| Models | 0.2 | | | | 0.7 | | | | 1 | | | |
| | Acc | $S_{GT}$ | Con | $S_C$ | Acc | $S_{GT}$ | Con | $S_C$ | Acc | $S_{GT}$ | Con | $S_{Con}$ |
| BLIP-2 | 25.1 | 52.9 | 48.2 | 63.9 | 25.1 | 52.9 | 48.2 | 63.9 | 25.1 | 52.9 | 48.2 | 63.9 |
| mPLUG-Owl2 | 33.2 | 63.8 | 48.6 | 63.7 | 30.4 | 61.1 | 36.7 | 55.6 | 27.7 | 58.1 | 30.7 | 51.3 |
| LLaVa 1.5M | 31.2 | 62.6 | 46.0 | 62.4 | 26.9 | 59.2 | 32.5 | 53.8 | 19.3 | 19.3 | 34.3 | 62.3 |
| MoE-LLaVa | 28.8 | 58.0 | 45.3 | 61.2 | 26.3 | 56.9 | 34.8 | 54.7 | 24.2 | 54.5 | 26.4 | 49.4 |
| Qwen-VL-Chat | 36.7 | 70.6 | 56.4 | 70.4 | 36.3 | 70.2 | 55.3 | 69.7 | 36.3 | 70.2 | 55.3 | 69.7 |
| BLIP-3 | 30.9 | 61.0 | 47.7 | 64.0 | 30.9 | 61.0 | 47.7 | 64.0 | 30.9 | 61.0 | 47.7 | 64.0 |

| Image Restyling Task | | | | | | | | | | | | |
|---|---|---|---|---|---|---|---|---|---|---|---|---|
| Models | 0.2 | | | | 0.7 | | | | 1 | | | |
| | Acc | $S_{GT}$ | Con | $S_C$ | Acc | $S_{GT}$ | Con | $S_C$ | Acc | $S_{GT}$ | Con | $S_{Con}$ |
| BLIP-2 | 13.0 | 17.0 | 38.4 | 62.8 | 13.0 | 17.0 | 38.4 | 62.8 | 13.0 | 17.0 | 38.4 | 62.8 |
| mPLUG-Owl2 | 9.6 | 15.4 | 46.1 | 66.6 | 9.0 | 15.2 | 26.3 | 59.2 | 8.4 | 14.9 | 13.5 | 53.8 |
| LLaVa 1.5M | 10.3 | 15.5 | 44.4 | 66.0 | 9.6 | 14.9 | 19.0 | 56.9 | 10.5 | 15.5 | 50.1 | 68.5 |
| MoE-LLaVa | 12.9 | 16.6 | 70.3 | 74.8 | 11.9 | 16.2 | 48.0 | 67.5 | 11.7 | 15.7 | 28.2 | 62.0 |
| Qwen-VL-Chat | 8.4 | 15.8 | 24.4 | 54.1 | 8.3 | 15.7 | 23.1 | 53.3 | 8.3 | 15.7 | 23.1 | 53.3 |
| BLIP-3 | 11.9 | 17.3 | 51.4 | 68.2 | 11.9 | 17.3 | 51.4 | 68.2 | 11.9 | 17.3 | 51.4 | 68.2 |

| Context Reasoning Task | | | | | | | | | | | | |
|---|---|---|---|---|---|---|---|---|---|---|---|---|
| Models | 0.2 | | | | 0.7 | | | | 1 | | | |
| | Acc | $S_{GT}$ | Con | $S_C$ | Acc | $S_{GT}$ | Con | $S_C$ | Acc | $S_{GT}$ | Con | $S_{Con}$ |
| BLIP-2 | 27.9 | 39.0 | 825 | 88.8 | 27.9 | 39.0 | 82.5 | 88.8 | 27.9 | 39.0 | 82.4 | 88.8 |
| mPLUG-Owl2 | 27.4 | 41.0 | 55.4 | 67.0 | 24.5 | 35.3 | 27.6 | 48.0 | 22.1 | 30.1 | 10.5 | 37.2 |
| LLaVa 1.5M | 28.2 | 42.0 | 62.4 | 72.4 | 20.1 | 28.9 | 25.9 | 42.3 | 22.9 | 37.8 | 15.4 | 41.6 |
| MoE-LLaVa | 25.2 | 38.5 | 39.4 | 56.9 | 24.4 | 34.7 | 22.0 | 45.1 | 23.8 | 31.6 | 10.0 | 38.7 |
| Qwen-VL-Chat | 20.4 | 32.2 | 32.9 | 50.6 | 20.1 | 32.2 | 30.7 | 49.0 | 20.1 | 32.2 | 30.7 | 49.0 |
| BLIP-3 | 28.0 | 36.5 | 40.0 | 54.9 | 28.0 | 36.5 | 40.0 | 54.9 | 28.0 | 36.5 | 40.0 | 54.9 |

### A.8 MORE ANALYSIS

We provide a deeper analysis of the Image Restyling and Context Reasoning tasks. In the Image Restyling task, we assess performance across 4 styles: Candy, Mosaic, Udnie, and Grayscale. According to Table 10, all MLLMs achieve their best results with the `Grayscale` transformation, which is expected as this style minimally changes the original image. Conversely, the `Mosaic` style proves challenging for all models, likely due to its significant modification of object appearances, leading to potential confusion for the MLLMs.

In the context reasoning task, we assess performance based on masking colors, lines, and shapes. Table 11 displays the results for various masking colors, with `Black` outperforming other colors. This could be attributed to the frequent use of black as a bounding box or mask in existing datasets, making MLLMs more adept at handling black masks. Table 12 examines the effect of different numbers of masking lines, where masks with three lines perform the best, in contrast to those with only one line. This difference may arise from the area covered by the masks or the additional cues multiple lines provide about the underlying objects, aiding MLLMs in making predictions. According to Table 13, open-source MLLMs show a preference for `Ellipse` masks over Rectangles, potentially due to the smaller coverage area of ellipses, similar to the influence of the number of lines.

From these observations, it appears that mask color has a minimal impact on performance. Instead, the area covered by the mask plays a more crucial role, suggesting that the inferencing capability of MLLMs could be further improved by addressing their sensitivity to the extent of occlusion.

### A.9 ADDITIONAL QUALITATIVE RESULTS

#### A.9.1 QUESTION REPHRASING

We show some qualitative results of the question rephrasing task in Figure 7. The closed-source models outperform other models in terms of both accuracy and consistency. While the accuracy of the evaluated MLLMs may not meet high standards, the similarity between the generated responses and the ground truth maintains a satisfactory level ($S_{GT}$), suggesting that the responses are semantically aligned with the ground truth. Regarding consistency, the majority of MLLMs demonstrate the capability to generate semantically similar answers to rephrased questions. In the models (*e.g.* LLaVa 1.5M (Improved) and BLIP2 (Improved)), the consistency improves significantly, showing the effectiveness of the proposed adapter.

#### A.9.2 IMAGE RESTYLING

The qualitative results of the image restyling task are depicted in Figure 8 and Figure 9. In this task, the BLIP-2 model outperforms other MLLMs, particularly in terms of consistency. Although LLaVa 1.5M and MoE-LLaVa may not always provide responses that align perfectly with the ground truth, their answers remain semantically consistent across various styles. This consistency is evident in their strong performance on metrics such as Consistency and $S_C$.

#### A.9.3 CONTEXT REASONING

Figure 10 and Figure 11 shows additional examples from the context reasoning task, where most MLLMs generally yield similar responses as the ground truth. However, it is observed that models like Owen-VL, MoE-LLaVa, and BLIP-3 are more influenced by the presence of masks, often incorporating the mask's color or shape into their answers. Another noteworthy trend is GPT-4V's tendency to respond with "I cannot provide..." when the masks obscure a significant portion of the objects, indicating a threshold of visual information required for it to generate confident responses.

Table 10: Results across different styles in the **Image Restyling Task**.

| | Models | | Candy | | | | Mosaic | | |
| --- | --- | --- | --- | --- | --- | --- | --- | --- | --- |
| | | Acc | $S_{GT}$ | Con | $S_C$ | Acc | $S_{GT}$ | Con | $S_C$ |
| All | BLIP-2 | 13.59 | 16.73 | 40.99 | 63.80 | 10.51 | 16.24 | 48.35 | 67.91 |
| | mPLUG-Owl2 | 8.33 | 15.79 | 24.17 | 58.40 | 4.28 | 14.81 | 29.65 | 60.86 |
| | LLaVa 1.5M | 10.21 | 15.76 | 52.55 | 69.04 | 7.51 | 15.53 | 55.78 | 70.79 |
| | MoE-LLaVa | 13.59 | 17.21 | 69.74 | 74.45 | 9.08 | 17.00 | 74.70 | 76.47 |
| | Qwen-VL-Chat | 7.51 | 15.88 | 20.27 | 51.69 | 3.45 | 15.00 | 29.35 | 57.96 |
| | BLIP-3 | 12.84 | 17.66 | 56.31 | 70.15 | 7.81 | 17.60 | 58.41 | 71.48 |
| | Gemini | 14.11 | 16.58 | 51.88 | 68.63 | 10.21 | 16.50 | 59.01 | 70.75 |
| | GPT-4V | 11.11 | 16.91 | 57.64 | 67.58 | 7.64 | 15.70 | 59.03 | 69.05 |
| | GPT-4o | 15.84 | 17.55 | 81.23 | 77.61 | 12.69 | 17.15 | 86.56 | 79.31 |
| Failure Cases | BLIP-2 | - | 16.59 | 39.90 | 63.04 | - | 16.11 | 46.55 | 66.98 |
| | mPLUG-Owl2 | - | 15.76 | 23.50 | 58.09 | - | 14.78 | 28.73 | 60.55 |
| | LLaVa 1.5M | - | 15.71 | 51.27 | 68.48 | - | 15.48 | 54.05 | 70.11 |
| | MoE-LLaVa | - | 17.23 | 68.74 | 74.03 | - | 16.97 | 73.51 | 76.13 |
| | Qwen-VL-Chat | - | 15.91 | 19.59 | 51.28 | - | 15.00 | 28.80 | 57.61 |
| | BLIP-3 | - | 17.69 | 55.52 | 69.81 | - | 17.57 | 58.62 | 71.30 |
| | Gemini | - | 16.58 | 50.16 | 68.20 | - | 16.43 | 57.17 | 70.25 |
| | GPT-4V | - | 16.96 | 56.43 | 67.18 | - | 15.68 | 57.86 | 68.57 |
| | GPT-4o | | 17.51 | 79.52 | 77.05 | - | 17.02 | 85.63 | 78.93 |

| | Models | | Udnie | | | | Grayscale | | |
| --- | --- | --- | --- | --- | --- | --- | --- | --- | --- |
| | | Acc | $S_{GT}$ | Con | $S_C$ | Acc | $S_{GT}$ | Con | $S_C$ |
| All | BLIP-2 | 10.96 | 16.78 | 36.94 | 62.32 | 16.97 | 18.32 | 40.69 | 63.71 |
| | mPLUG-Owl2 | 7.96 | 14.99 | 29.43 | 60.53 | 15.24 | 15.41 | 21.92 | 57.31 |
| | LLaVa 1.5M | 9.01 | 15.36 | 46.17 | 67.11 | 15.17 | 15.32 | 51.20 | 69.87 |
| | MoE-LLaVa | 12.31 | 16.24 | 68.99 | 74.43 | 16.74 | 15.95 | 70.42 | 74.31 |
| | Qwen-VL-Chat | 7.43 | 15.45 | 22.37 | 54.13 | 14.71 | 16.57 | 20.95 | 50.74 |
| | BLIP-3 | 10.29 | 16.50 | 47.52 | 67.19 | 16.74 | 17.47 | 52.63 | 68.06 |
| | Gemini | 13.21 | 15.98 | 55.56 | 69.66 | 17.19 | 15.54 | 50.83 | 68.70 |
| | GPT-4V | 6.25 | 15.59 | 47.22 | 66.38 | 13.89 | 15.38 | 52.08 | 66.11 |
| | GPT-4o | 16.59 | 16.53 | 77.40 | 76.09 | 20.12 | 16.09 | 79.80 | 77.22 |
| Failure Cases | BLIP-2 | - | 16.64 | 35.77 | 61.53 | - | 18.15 | 39.42 | 62.84 |
| | mPLUG-Owl2 | - | 14.97 | 28.65 | 60.22 | - | 15.40 | 20.89 | 56.96 |
| | LLaVa 1.5M | - | 15.29 | 44.60 | 66.43 | - | 15.24 | 49.84 | 69.31 |
| | MoE-LLaVa | - | 16.27 | 67.61 | 74.06 | - | 15.99 | 69.47 | 74.06 |
| | Qwen-VL-Chat | - | 15.43 | 21.66 | 53.79 | - | 16.58 | 20.43 | 50.35 |
| | BLIP-3 | - | 16.49 | 46.47 | 66.86 | - | 17.43 | 51.95 | 67.75 |
| | Gemini | - | 15.95 | 53.70 | 69.19 | - | 15.57 | 49.68 | 68.36 |
| | GPT-4V | - | 15.58 | 45.71 | 66.06 | - | 15.36 | 50.71 | 65.59 |
| | GPT-4o | - | 16.50 | 75.64 | 75.57 | - | 16.10 | 78.78 | 76.94 |

Table 11: Results across different masking color in the **Context Reasoning Task**.

| | Models | Blue Acc | $S_{GT}$ | Con | $S_C$ | Red Acc | $S_{GT}$ | Con | $S_C$ |
|---|---|---|---|---|---|---|---|---|---|
| All | BLIP-2 | 26.67 | 38.27 | 82.70 | 89.50 | 27.88 | 39.09 | 79.89 | 87.71 |
| | mPLUG-Owl2 | 23.51 | 34.62 | 26.07 | 46.06 | 23.06 | 35.26 | 27.88 | 48.38 |
| | LLaVa 1.5M | 26.31 | 41.51 | 69.18 | 77.44 | 27.88 | 42.22 | 65.82 | 75.70 |
| | MoE-LLaVa | 20.34 | 35.60 | 37.27 | 56.10 | 23.06 | 37.16 | 35.79 | 54.81 |
| | Qwen-VL-Chat | 19.85 | 31.78 | 33.50 | 50.14 | 17.69 | 29.97 | 28.95 | 47.80 |
| | BLIP-3 | 23.75 | 35.44 | 39.71 | 54.30 | 28.02 | 37.93 | 40.08 | 54.84 |
| | Gemini | 54.95 | 56.52 | 44.81 | 62.90 | 51.54 | 54.42 | 43.89 | 61.95 |
| | GPT-4V | 32.22 | 20.39 | 30.56 | 56.53 | 28.83 | 18.83 | 33.74 | 56.90 |
| | GPT-4o | 50.06 | 31.26 | 48.11 | 66.51 | 52.41 | 31.80 | 47.05 | 66.06 |
| Failure Cases | BLIP-2 | - | 27.56 | 78.04 | 87.26 | - | 29.21 | 74.79 | 85.05 |
| | mPLUG-Owl2 | - | 30.55 | 21.21 | 42.66 | - | 30.95 | 24.12 | 45.59 |
| | LLaVa 1.5M | - | 32.49 | 64.05 | 73.73 | - | 33.10 | 59.93 | 71.72 |
| | MoE-LLaVa | - | 30.65 | 31.63 | 52.76 | - | 32.01 | 30.04 | 51.16 |
| | Qwen-VL-Chat | - | 27.21 | 28.76 | 46.66 | - | 26.11 | 24.17 | 44.60 |
| | BLIP-3 | - | 30.47 | 35.61 | 50.71 | - | 32.45 | 34.56 | 50.19 |
| | Gemini | - | 45.78 | 28.33 | 51.84 | - | 44.08 | 30.48 | 52.99 |
| | GPT-4V | - | 17.09 | 26.11 | 54.20 | - | 13.54 | 28.26 | 54.14 |
| | GPT-4o | - | 25.15 | 41.40 | 64.35 | - | 24.66 | 39.52 | 63.38 |

| | Models | Green Acc | $S_{GT}$ | Con | $S_C$ | Yellow Acc | $S_{GT}$ | Con | $S_C$ |
|---|---|---|---|---|---|---|---|---|---|
| All | BLIP-2 | 27.76 | 39.54 | 85.44 | 90.36 | 28.23 | 38.64 | 83.60 | 89.46 |
| | mPLUG-Owl2 | 23.13 | 34.06 | 24.90 | 45.90 | 25.00 | 35.72 | 28.23 | 48.94 |
| | LLaVa 1.5M | 28.30 | 42.14 | 67.48 | 76.50 | 26.88 | 41.57 | 70.03 | 78.57 |
| | MoE-LLaVa | 22.99 | 36.26 | 39.46 | 56.35 | 19.89 | 34.29 | 39.65 | 57.32 |
| | Qwen-VL-Chat | 18.78 | 31.92 | 30.20 | 48.70 | 18.68 | 30.69 | 32.12 | 50.19 |
| | BLIP-3 | 23.67 | 36.12 | 41.90 | 55.92 | 24.19 | 34.22 | 38.58 | 54.32 |
| | Gemini | 57.96 | 57.28 | 45.71 | 62.09 | 52.76 | 55.19 | 45.76 | 63.16 |
| | GPT-4V | 30.49 | 21.32 | 37.20 | 57.86 | 33.11 | 21.01 | 38.41 | 59.81 |
| | GPT-4o | 50.88 | 31.49 | 51.70 | 67.16 | 51.21 | 31.64 | 50.54 | 67.32 |
| Failure Cases | BLIP-2 | - | 29.78 | 81.63 | 88.30 | - | 27.43 | 78.70 | 87.08 |
| | mPLUG-Owl2 | - | 32.47 | 61.49 | 72.54 | - | 31.96 | 64.85 | 75.02 |
| | LLaVa 1.5M | - | 30.20 | 20.66 | 42.89 | - | 30.49 | 23.30 | 45.24 |
| | MoE-LLaVa | - | 31.10 | 33.84 | 52.82 | - | 28.41 | 33.83 | 53.69 |
| | Qwen-VL-Chat | - | 28.28 | 26.69 | 45.92 | - | 26.51 | 26.93 | 46.65 |
| | BLIP-3 | - | 30.16 | 36.70 | 51.96 | - | 28.39 | 32.72 | 50.30 |
| | Gemini | - | 44.87 | 25.27 | 47.69 | - | 43.47 | 30.10 | 52.20 |
| | GPT-4V | - | 16.64 | 33.09 | 55.59 | - | 18.47 | 35.82 | 58.54 |
| | GPT-4o | - | 24.65 | 44.06 | 64.39 | - | 24.43 | 43.00 | 64.77 |

| | Models | White Acc | $S_{GT}$ | Con | $S_C$ | Black Acc | $S_{GT}$ | Con | $S_C$ |
|---|---|---|---|---|---|---|---|---|---|
| All | BLIP-2 | 27.40 | 38.14 | 81.92 | 88.02 | 29.59 | 40.14 | 81.11 | 87.64 |
| | mPLUG-Owl2 | 24.68 | 34.62 | 30.13 | 49.65 | 27.48 | 37.71 | 28.93 | 49.13 |
| | LLaVa 1.5M | 27.98 | 42.03 | 68.01 | 76.96 | 34.74 | 45.64 | 67.64 | 76.87 |
| | MoE-LLaVa | 30.42 | 41.84 | 41.61 | 58.18 | 34.87 | 46.00 | 42.93 | 58.54 |
| | Qwen-VL-Chat | 21.66 | 33.11 | 31.85 | 49.50 | 24.04 | 35.80 | 27.34 | 47.30 |
| | BLIP-3 | 31.13 | 37.06 | 41.46 | 56.17 | 37.38 | 38.22 | 38.57 | 53.86 |
| | Gemini | 57.10 | 56.54 | 44.62 | 62.31 | 57.09 | 57.53 | 46.23 | 63.04 |
| | GPT-4V | 30.20 | 21.81 | 37.58 | 57.89 | 35.95 | 22.86 | 31.37 | 55.80 |
| | GPT-4o | 53.66 | 32.04 | 50.22 | 67.35 | 52.44 | 31.84 | 49.54 | 66.58 |
| Failure Cases | BLIP-2 | - | 28.85 | 77.26 | 85.57 | - | 29.33 | 76.30 | 85.14 |
| | mPLUG-Owl2 | - | 30.26 | 25.60 | 46.48 | - | 33.81 | 25.18 | 46.32 |
| | LLaVa 1.5M | - | 32.58 | 62.14 | 73.10 | - | 37.15 | 61.33 | 72.64 |
| | MoE-LLaVa | - | 36.67 | 35.17 | 53.95 | - | 40.16 | 36.20 | 53.80 |
| | Qwen-VL-Chat | - | 29.05 | 27.30 | 46.17 | - | 32.33 | 23.10 | 44.20 |
| | BLIP-3 | - | 32.44 | 35.86 | 51.84 | - | 33.58 | 31.55 | 48.78 |
| | Gemini | - | 46.09 | 28.35 | 50.30 | - | 46.44 | 28.63 | 50.75 |
| | GPT-4V | - | 18.44 | 32.58 | 55.81 | - | 18.44 | 25.78 | 53.19 |
| | GPT-4o | - | 26.26 | 40.13 | 64.92 | - | 24.94 | 42.16 | 63.82 |

Table 12: Results across different numbers of lines in the **Context Reasoning Task**.

|  | Models | Acc | $S_{GT}$ | Con | $S_C$ | Acc | $S_{GT}$ | Con | $S_C$ |
|---|---|---|---|---|---|---|---|---|---|
|  |  | **1** | | | | **3** | | | |
| All | BLIP-2 | 23.71 | 36.08 | 78.44 | 86.34 | 32.10 | 41.27 | 82.75 | 88.91 |
|  | mPLUG-Owl2 | 19.50 | 31.66 | 25.09 | 45.49 | 29.43 | 38.16 | 30.17 | 48.93 |
|  | LLaVa 1.5M | 23.88 | 38.60 | 61.86 | 72.55 | 33.76 | 46.20 | 69.83 | 78.16 |
|  | MoE-LLaVa | 17.27 | 32.56 | 32.99 | 53.31 | 30.54 | 42.35 | 42.99 | 58.78 |
|  | Qwen-VL-Chat | 13.06 | 25.69 | 26.37 | 45.45 | 28.78 | 37.72 | 33.95 | 50.83 |
|  | BLIP-3 | 15.81 | 30.66 | 34.19 | 50.24 | 37.82 | 41.47 | 43.82 | 58.42 |
|  | Gemini | 55.98 | 56.17 | 42.73 | 61.62 | 54.24 | 56.60 | 47.51 | 63.40 |
|  | GPT-4V | 35.25 | 20.33 | 32.38 | 55.62 | 33.33 | 21.63 | 34.17 | 58.25 |
|  | GPT-4o | 46.65 | 29.32 | 48.88 | 66.38 | 56.83 | 33.65 | 49.91 | 66.88 |
| Failure Cases | BLIP-2 | - | 26.47 | 73.36 | 83.62 | - | 30.24 | 77.53 | 86.15 |
|  | mPLUG-Owl2 | - | 27.70 | 20.96 | 42.48 | - | 33.19 | 25.05 | 45.25 |
|  | LLaVa 1.5M | - | 29.90 | 55.95 | 68.54 | - | 36.36 | 63.39 | 73.96 |
|  | MoE-LLaVa | - | 27.70 | 27.32 | 49.79 | - | 36.17 | 36.06 | 54.40 |
|  | Qwen-VL-Chat | - | 22.18 | 22.15 | 42.44 | - | 32.94 | 28.18 | 46.68 |
|  | BLIP-3 | - | 26.06 | 29.45 | 46.74 | - | 35.58 | 37.83 | 53.51 |
|  | Gemini | - | 45.94 | 25.78 | 50.20 | - | 44.38 | 29.12 | 50.80 |
|  | GPT-4V | - | 15.74 | 27.67 | 52.91 | - | 18.82 | 30.70 | 56.64 |
|  | GPT-4o | - | 22.48 | 40.88 | 63.66 | - | 26.96 | 40.42 | 64.06 |
|  |  | **5** | | | | **7** | | | |
| All | BLIP-2 | 29.90 | 40.23 | 84.28 | 89.98 | 26.29 | 38.51 | 84.45 | 90.05 |
|  | mPLUG-Owl2 | 26.02 | 36.44 | 26.92 | 48.63 | 23.32 | 35.34 | 28.56 | 48.92 |
|  | LLaVa 1.5M | 30.08 | 43.75 | 70.82 | 78.99 | 27.34 | 41.81 | 69.96 | 78.56 |
|  | MoE-LLaVa | 28.09 | 40.00 | 39.93 | 57.33 | 25.24 | 39.32 | 42.01 | 58.20 |
|  | Qwen-VL-Chat | 19.87 | 33.27 | 32.25 | 50.75 | 19.30 | 32.57 | 30.48 | 48.98 |
|  | BLIP-3 | 31.53 | 37.85 | 40.38 | 54.29 | 27.51 | 36.35 | 42.01 | 56.80 |
|  | Gemini | 58.42 | 57.86 | 46.11 | 63.49 | 52.23 | 54.45 | 44.53 | 61.93 |
|  | GPT-4V | 31.19 | 21.95 | 35.32 | 57.87 | 27.52 | 20.22 | 36.82 | 58.01 |
|  | GPT-4o | 54.56 | 33.07 | 49.32 | 67.00 | 49.34 | 30.81 | 49.87 | 67.03 |
| Failure Cases | BLIP-2 | - | 30.10 | 79.95 | 87.98 | - | 28.21 | 80.52 | 88.05 |
|  | mPLUG-Owl2 | - | 32.55 | 23.18 | 45.74 | - | 31.04 | 24.13 | 45.87 |
|  | LLaVa 1.5M | - | 34.23 | 65.65 | 75.23 | - | 33.06 | 64.82 | 75.15 |
|  | MoE-LLaVa | - | 34.53 | 34.01 | 53.50 | - | 34.27 | 36.55 | 54.58 |
|  | Qwen-VL-Chat | - | 29.47 | 28.03 | 47.85 | - | 28.91 | 26.67 | 46.08 |
|  | BLIP-3 | - | 32.71 | 34.37 | 49.77 | - | 31.24 | 36.89 | 52.85 |
|  | Gemini | - | 47.07 | 30.64 | 52.25 | - | 43.07 | 29.02 | 51.01 |
|  | GPT-4V | - | 17.66 | 30.16 | 55.79 | - | 16.08 | 32.11 | 55.54 |
|  | GPT-4o | - | 26.55 | 43.16 | 64.69 | - | 24.40 | 42.44 | 64.70 |

Table 13: Results across different masking shapes in the **Context Reasoning Task**.

|  | Models | Acc | $S_{GT}$ | Con | $S_C$ | Acc | $S_{GT}$ | Con | $S_C$ |
|---|---|---|---|---|---|---|---|---|---|
|  |  | **Rectangle** | | | | **Ellipse** | | | |
| All | BLIP-2 | 26.51 | 38.32 | 82.66 | 88.98 | 29.22 | 39.58 | 82.24 | 88.63 |
|  | mPLUG-Owl2 | 22.34 | 34.12 | 27.39 | 48.01 | 26.47 | 36.47 | 27.89 | 47.93 |
|  | LLaVa 1.5M | 26.42 | 41.49 | 67.84 | 77.24 | 30.78 | 43.48 | 68.23 | 76.80 |
|  | MoE-LLaVa | 20.28 | 35.60 | 39.59 | 56.91 | 29.74 | 41.17 | 39.22 | 56.81 |
|  | Qwen-VL-Chat | 16.33 | 29.24 | 27.89 | 47.01 | 23.66 | 34.99 | 33.32 | 50.77 |
|  | BLIP-3 | 24.13 | 35.57 | 40.83 | 55.29 | 31.55 | 37.34 | 39.27 | 54.48 |
|  | Gemini | 54.39 | 56.20 | 45.06 | 62.34 | 55.98 | 56.30 | 45.27 | 62.8 |
|  | GPT-4V | 33.19 | 21.16 | 34.48 | 57.16 | 30.43 | 20.83 | 34.89 | 57.69 |
|  | GPT-4o | 47.71 | 30.18 | 50.46 | 67.07 | 55.52 | 33.07 | 48.58 | 66.58 |
| Failure Cases | BLIP-2 | - | 27.96 | 78.13 | 86.64 | - | 29.36 | 77.45 | 86.19 |
|  | mPLUG-Owl2 | - | 29.94 | 23.06 | 45.00 | - | 32.10 | 23.50 | 44.63 |
|  | LLaVa 1.5M | - | 32.47 | 62.37 | 73.58 | - | 34.05 | 62.31 | 72.71 |
|  | MoE-LLaVa | - | 30.28 | 33.86 | 53.35 | - | 35.71 | 32.93 | 52.68 |
|  | Qwen-VL-Chat | - | 26.04 | 24.26 | 44.36 | - | 30.36 | 28.02 | 47.01 |
|  | BLIP-3 | - | 30.24 | 35.64 | 51.27 | - | 32.16 | 33.42 | 49.99 |
|  | Gemini | - | 45.49 | 28.64 | 51.24 | - | 44.72 | 28.54 | 50.87 |
|  | GPT-4V | - | 16.56 | 29.37 | 54.58 | - | 17.53 | 30.95 | 55.82 |
|  | GPT-4o | - | 23.60 | 42.97 | 64.80 | - | 26.39 | 40.48 | 63.75 |

**Prompt**: Please answer the question with a very-short answer. + <**R1/R2/R3**>
**R1)** What material was utilized to create the vessels shown in the picture?
**R2)** What substance was used in the production of the vessels featured in the picture?
**R3)** Can you identify the material from which the vessels in the picture were made?

**Q)** What is the material used to make the vessels in this picture?

**A)** copper

| LLaVa 1.5M | BLIP-2 | Qwen-VL | LLaVa 1.5M (Improved) | BLIP-2 (Improved) |
|---|---|---|---|---|
| **A1)** copper | **A1)** stainless steel | **A1)** copper | **A1)** copper | **A1)** stainless steel |
| **A2)** copper | **A2)** copper | **A2)** copper | **A2)** copper | **A2)** stainless steel |
| **A3)** yes | **A3)** stainless steel | **A3)** copper | **A3)** copper | **A3)** stainless steel |
| Acc: 66.6 | Acc: 33.3 | Acc: 100 | Acc: 100 | Acc: 0 |
| $S_{GT}$: 70.4 | $S_{GT}$: 67.2 | $S_{GT}$: 100 | $S_{GT}$: 100 | $S_{GT}$: 50.9 |
| Con: 33.3 | Con: 33.3 | Con: 100 | Con: 100 | Con: 100 |
| $S_C$: 67.2 | $S_C$: 67.2 | $S_C$: 100 | $S_C$: 100 | $S_C$: 100 |

| mPlug_owl2 | moellava | BLIP-3 | GPT-4V | GPT-4o | Gemini |
|---|---|---|---|---|---|
| **A1)** copper | **A1)** metal | **A1)** copper | **A1)** copper | **A1)** copper | **A1)** copper |
| **A2)** copper | **A2)** copper | **A2)** copper | **A2)** copper | **A2)** copper | **A2)** copper |
| **A3)** yes | **A3)** yes | **A3)** yes | **A3)** copper | **A3)** copper | **A3)** copper |
| Acc: 66.6 | Acc: 33.3 | Acc: 66.6 | Acc: 100 | Acc: 100 | Acc: 100 |
| $S_{GT}$: 70.4 | $S_{GT}$: 51.6 | $S_{GT}$: 70.4 | $S_{GT}$: 100 | $S_{GT}$: 100 | $S_{GT}$: 100 |
| Con: 33.3 | Con: 0 | Con: 33.3 | Con: 100 | Con: 100 | Con: 100 |
| $S_C$: 67.2 | $S_C$: 28.3 | $S_C$: 67.2 | $S_C$: 100 | $S_C$: 100 | $S_C$: 100 |

**Prompt**: Please answer the question with a very-short answer. + <**R1/R2/R3**>
**R1)** What is the designated name for the location where these benches are found?
**R2)** In which specific park can these benches be found?
**R3)** What is the official name of the park where these benches reside?

**Q)** What is the name of the park these benches reside in?

**A)** central park

| LLaVa 1.5M | BLIP-2 | mPlug_owl2 | moellava |
|---|---|---|---|
| **A1)** park | **A1)** park | **A1)** park | **A1)** park |
| **A2)** park | **A2)** sydney harbour foreshore park | **A2)** Central park | **A2)** Central park |
| **A3)** unknown | **A3)** East River Promenade Park | **A3)** Central park | **A3)** Stanley park |
| Acc: 0 | Acc: 0 | Acc: 66.6 | Acc: 33.3 |
| $S_{GT}$: 59.0 | $S_{GT}$: 56.0 | $S_{GT}$: 91.8 | $S_{GT}$: 82.0 |
| Con: 33.3 | Con: 0 | Con: 100 | Con: 100 |
| $S_C$: 50.3 | $S_C$: 47.0 | $S_C$: 83.6 | $S_C$: 73.6 |

| Qwen-VL | BLIP-3 | GPT-4V |
|---|---|---|
| **A1)** park | **A1)** park | **A1)** park |
| **A2)** unanswerable | **A2)** Central park | **A2)** I don't know where these benches can be found. |
| **A3)** unanswerable | **A3)** Central park | **A3)** I don't know the name of the park. |
| Acc: 0 | Acc: 66.6 | Acc: 0 |
| $S_{GT}$: 33.9 | $S_{GT}$: 91.8 | $S_{GT}$: 56.4 |
| Con: 33.3 | Con: 100 | Con: 0 |
| $S_C$: 47.5 | $S_C$: 83.6 | $S_C$: 45.4 |

| GPT-4o | Gemini | LLaVa 1.5M (Improved) | BLIP-2 (Improved) |
|---|---|---|---|
| **A1)** park | **A1)** Waterfront Park | **A1)** park | **A1)** park |
| **A2)** Stanley Park | **A2)** Stanley Park | **A2)** park | **A2)** park |
| **A3)** I don't know | **A3)** Stanley Park | **A3)** park | **A3)** park |
| Acc: 0 | Acc: 0 | Acc: 0 | Acc: 0 |
| $S_{GT}$: 53.2 | $S_{GT}$: 66.7 | $S_{GT}$: 75.4 | $S_{GT}$: 75.4 |
| Con: 33.3 | Con: 33.3 | Con: 100 | Con: 100 |
| $S_C$: 31.4 | $S_C$: 77.4 | $S_C$: 100 | $S_C$: 100 |

Figure 7: Qualitative Results for the **Question Rephrasing Task**.

**Prompt**: Please describe the place in the image in two sentences.

**GT**: inside bus

**S1)** Candy

**S2)** Mosaic

**S3)** Udnie

**S4)** Grayscale

**BLIP-2**
**S1)** the inside of a bus
**S2)** the inside of a bus with stained glass windows
**S3)** the interior of a bus
**S4)** the inside of a bus

Acc: 100
$S_{GT}$: 81.7
Con: 50
$S_C$: 82.2

**LLaVa 1.5M (Improved)**
**S1)** bus
**S2)** inside of a bus
**S3)** bus
**S4)** inside of a bus

Acc: 50
$S_{GT}$: 88.3
Con: 100
$S_C$: 85.1

**BLIP-2 (Improved)**
**S1)** inside bus
**S2)** inside bus
**S3)** inside bus
**S4)** inside bus

Acc: 100
$S_{GT}$: 100
Con: 100
$S_C$: 100

**LLaVa 1.5M**
**S1)** A bus with a yellow and red interior is shown in the image.
**S2)** The image features a bus with a colorful and artistic design on the seats.
**S3)** A bus with a row of seats and a window.
**S4)** The image shows the interior of a bus with a row of seats and a window.

Acc: 0
$S_{GT}$: 60.3
Con: 66.7
$S_C$: 73.7

**mPlug_owl2**
**S1)** The image shows a bus with a colorful interior, featuring various seats and a mirror on the wall.
**S2)** The image shows a colorful glass mosaic depicting a stained-glass train car with a colorful design, likely showcasing the train seats and windows.
**S3)** The image is a drawing of a passenger bus with various seating arrangements, including benches and chairs.
**S4)** The image shows a bus with empty seats and a blank space in the center of the front area.

Acc: 0
$S_{GT}$: 45.6
Con: 16.7
$S_C$: 57.5

**moellava**
**S1)** The image shows the inside of a bus, with a row of seats and a window. The seats are colorful and arranged in rows.
**S2)** The image shows the interior of a bus, with a colorful and artistic design. The seats are arranged in rows, and the overall atmosphere is vibrant and visually appealing.
**S3)** The image shows the interior of a bus with a row of seats, some of which are empty. The seats are covered with a blue and orange striped fabric.
**S4)** The image shows the interior of a bus, with rows of seats and a window. The bus is empty, and the seats are arranged in a neat and orderly fashion.

Acc: 25
$S_{GT}$: 62.2
Con: 100
$S_C$: 86.3

**Qwen-VL**
**S1)** A bus with a yellow and blue interior and red seats.
**S2)** A stained glass window showing a car with seats for five people.
**S3)** The inside of a bus with blue and grey seats.
**S4)** An empty bus with many seats and windows.

Acc: 0
$S_{GT}$: 53.1
Con: 16.7
$S_C$: 51.9

**BLIP-3**
**S1)** The image is a painting of the inside of a bus. The seats are red and blue and there are two speakers on the ceiling.
**S2)** The image depicts the interior of a bus, with rows of seats and windows. The seats are arranged in a grid-like pattern and the windows are rectangular, allowing light to filter into the bus.
**S3)** The interior of a bus with rows of seats.
**S4)** The image is a black and white photo of the inside of a bus. The bus is empty and has rows of seats on both sides of a central aisle.

Acc: 25
$S_{GT}$: 61.2
Con: 50
$S_C$: 68.9

**Gemini**
**S1)** This is the inside of a school bus, likely a smaller one than what is typically used for school children. The seats are patterned and lined up in rows, with some seats appearing empty and others having personal belongings on them.
**S2)** The image shows the interior of a bus, with the seats on either side of the aisle visible. The seats are patterned with orange and blue lines, and the windows are covered with a stained glass effect.
**S3)** The image shows the inside of a bus with rows of seats facing each other. The seats are covered in a colorful patterned fabric.
**S4)** The image shows the inside of a bus with rows of seats facing each other. There are windows on either side of the bus and the ceiling is white with a vent in the middle.

Acc: 50
$S_{GT}$: 57.3
Con: 66.7
$S_C$: 72.4

**GPT-4V**
**S1)** The image depicts the interior of a brightly colored bus, featuring a unique design with patterned seats and warm colors. Large windows line the sides, allowing natural light to fill the space, creating an inviting atmosphere.
**S2)** The image features an abstract representation of the interior of a bus, characterized by vibrant, colorful patterns on the seats. Large windows line the sides, allowing light to wash over the intricately drawn design, creating a whimsical atmosphere.
**S3)** The image depicts the interior of a bus or van, featuring several rows of seats arranged in a linear fashion. The seating is designed with patterned upholstery and there are large windows visible along the sides, allowing for natural light to enter the space.
**S4)** The image depicts the interior of a bus or shuttle, characterized by a clean and organized seating arrangement. The seats are upholstered in a patterned fabric, and there are multiple windows along the sides allowing for light to enter the space.

Acc: 0
$S_{GT}$: 55.0
Con: 100
$S_C$: 84.4

**GPT-4o**
**S1)** The image depicts the interior of a small bus or shuttle, showcasing rows of colorful, patterned seats. Large windows on the sides provide natural light, creating a bright and vibrant atmosphere.
**S2)** The image depicts the interior of a bus or train with rows of colorful, patterned seats. The windows on the sides lend a bright and airy feel to the transportation environment.
**S3)** The image shows the interior of a small passenger bus or shuttle with multiple seats arranged in rows. The seats are upholstered with a patterned fabric featuring blue and orange designs, and windows line the sides of the vehicle.
**S4)** The image shows the inside of a bus with rows of seats facing each other. There are windows on either side of the bus and the ceiling is white with a vent in the middle.

Acc: 25
$S_{GT}$: 53.1
Con: 83.3
$S_C$: 78.2

Figure 8: Qualitative Results for the **Image Restyling Task**.

**Prompt**: Please describe the place in the image in two sentences.

**GT**: office

**S1)** Candy

**S2)** Mosaic

**S3)** Udnie

**S4)** Grayscale

**BLIP-2**
**S1)** a painting of an office with a desk and a chair
**S2)** an office with a large window
**S3)** an office with a large window
**S4)** this is a black and white photo of an office
Acc: 100
$S_{GT}$: 54.9
Con: 16.7
$S_C$: 58.7

**LLaVa 1.5M (Improved)**
**S1)** living room
**S2)** living room
**S3)** living room
**S4)** living room
Acc: 0
$S_{GT}$: 49.3
Con: 100
$S_C$: 100

**BLIP-2 (Improved)**
**S1)** office
**S2)** office
**S3)** office
**S4)** office
Acc: 100
$S_{GT}$: 100
Con: 100
$S_C$: 100

**LLaVa 1.5M**
**S1)** The image features a living room with a red desk, a chair, and a potted plant.
**S2)** The image features a living room with a couch, a desk, and a window.
**S3)** The image features a large office space with a desk, chair, and a potted plant.
**S4)** The image shows a large office space with a desk, chair, and computer setup.
Acc: 50
$S_{GT}$: 43.6
Con: 83.3
$S_C$: 77.0

**mPlug_owl2**
**S1)** The image features a vibrant hotel room with a red bed, a wooden chest, a couch, and a wall with a painting, giving it a colorful, unique atmosphere.
**S2)** The image features a room with a large window, a couch, and a chair. There is also a computer in the room.
**S3)** The image features a well-decorated room with a large desk and a large window, reflecting a beautiful view outside.
**S4)** The image features a spacious office room with large windows, a desk, a chair, a television, and a potted plant.
Acc: 25
$S_{GT}$: 32.9
Con: 50
$S_C$: 70.7

**moellava**
**S1)** The image shows a room with a desk, a chair, a computer, and a window. There are also potted plants and a vase in the room.
**S2)** The image is a colorful, artistic representation of a room with a desk, a chair, a computer, and a potted plant. The room has a window and is decorated with stained glass.
**S3)** The image depicts a clean and organized office space with a desk, chair, and computer. The office is well-lit with natural light from two windows, and there are potted plants on the desk, adding a touch of greenery to the room.
**S4)** The image is a black and white photo of a clean and organized office space. The office features a desk with a chair, a computer, and a potted plant.
Acc: 50
$S_{GT}$: 43.2
Con: 100
$S_C$: 81.6

**Qwen-VL**
**S1)** A red table sitting in a room with two large windows.
**S2)** A room with a bed, a computer, and two potted plants.
**S3)** A room with a bed, two chairs, a desk and two computer monitors.
**S4)** A black and white image of a corner office with a desk and chair.
Acc: 25
$S_{GT}$: 34.9
Con: 16.7
$S_C$: 47.4

**BLIP-3**
**S1)** A painting of an office with a desk and a chair.
**S2)** A room with a desk and a chair.
**S3)** A painting of an office with a desk and a chair.
**S4)** A black and white photo of an office with a desk, chair, and a computer.
Acc: 75
$S_{GT}$: 48.9
Con: 33.3
$S_C$: 73.4

**Gemini**
**S1)** The image shows a well-lit office with two desks, both with computers on them. The office appears to be in a high-rise building with a large window offering a view of a city skyline.
**S2)** This is an office space with two desks, computers, and a large window that overlooks a body of water. The room is decorated with a plant, and the image is stylized with a stained-glass effect.
**S3)** The image shows a well-lit office space with a large desk, two chairs, and two computers. The office is modern and has a large window overlooking a cityscape.
**S4)** This is a spacious office with large windows overlooking a cityscape. It has a large desk, two computer monitors, a comfy chair, and a potted plant.
Acc: 100
$S_{GT}$: 50.1
Con: 100
$S_C$: 80.2

**GPT-4V**
**S1)** The image depicts an interior space with a modern, cozy ambiance, featuring a red sofa and a cushioned armchair that invite relaxation. Large windows let in ample natural light, illuminating the room and a variety of indoor plants that add a touch of greenery to the warm-toned decor.
**S2)** The image appears to be an artistic, abstract representation of a bedroom with a bed in the center, a large window providing a view of the outside, and possibly a nightstand or desk with a laptop. The image is stylized with distinct, bold lines and a vibrant color palette that distorts the usual perspective, giving it a dream-like or surreal quality.
**S3)** The image features an interior space with a modern design, including large glass windows that provide a view of the exterior environment. The room is furnished with a desk on which there is a computer, a plant, and various other items, suggesting a workspace or office setting.
**S4)** The image shows a well-furnished office with a modern aesthetic, featuring a large desk, comfortable chair, and multiple computer monitors, indicative of a technology-focused workspace. Large windows provide a view of the exterior, suggesting the office is located in a high-rise building, and the presence of plants adds a touch of nature to the room.
Acc: 50
$S_{GT}$: 35
Con: 50
$S_C$: 70.7

**GPT-4o**
**S1)** The image appears to be an office space with large windows, featuring two desks equipped with computers and an ergonomic chair. There are potted plants adding a touch of greenery, contributing to a vibrant and professional atmosphere.
**S2)** The image depicts a vibrant and artistic rendition of an office space, complete with desks, computers, and large windows that let in ample natural light. The stained-glass effect adds a colorful and abstract element to the scene, enhancing the overall ambiance.
**S3)** The image depicts a modern office space with large windows that provide a view of the outdoors. The office is furnished with a desk, computer, chair, and houseplants, creating a professional yet comfortable environment.
**S4)** This is a spacious office with large windows overlooking a cityscape. It has a large desk, two computer monitors, a comfy chair, and a potted plant.
Acc: 100
$S_{GT}$: 47.3
Con: 66.7
$S_C$: 70.7

Figure 9: Qualitative Results for the **Image Restyling Task**.

**Prompt**: What kind of object is in the masked region?

**GT**: orange

M1

M2

M3

**BLIP-2**
A1) oranges
A2) oranges
A3) oranges
Acc: 100
S$_{GT}$: 81.9
Con: 100
S$_C$: 100

**LLaVa 1.5M (Improved)**
A1) orange
A2) orange
A3) orange
Acc: 100
S$_{GT}$: 100
Con: 100
S$_C$: 100

**BLIP-2 (Improved)**
A1) orange
A2) orange
A3) orange
Acc: 100
S$_{GT}$: 100
Con: 100
S$_C$: 100

**LLaVa 1.5M**
A1) There is an orange in the masked region.
A2) There is a square object in the masked region.
A3) There is a green square in the masked region.
Acc: 33.3
S$_{GT}$: 42.5
Con: 0
S$_C$: 47.4

**mPlug_owl2**
A1) In the masked region, there is an orange.
A2) The masked region is a square.
A3) The masked region contains green leaves or branches.
Acc: 33.3
S$_{GT}$: 36.4
Con: 0
S$_C$: 30.2

**moellava**
A1) There is an orange in the masked region of the image.
A2) The masked region features an orange tree with oranges on it.
A3) There is a green object in the masked region, which is a green box.
Acc: 66.7
S$_{GT}$: 53.7
Con: 0
S$_C$: 40.3

**Qwen-VL**
A1) The masked region contains an orange fruit hanging from a tree.
A2) The object in the masked region is an orange.
A3) The object in the masked region is a green square.
Acc: 66.7
S$_{GT}$: 50.0
Con: 0
S$_C$: 50.1

**BLIP-3**
A1) The masked region contains an orange.
A2) The masked region is a white rectangular shape with horizontal lines.
A3) The masked region is a green rectangle with horizontal lines.
Acc: 33.3
S$_{GT}$: 25.7
Con: 33.3
S$_C$: 65.2

**Gemini**
A1) The masked region contains an orange.
A2) The masked region contains a fruit. It is likely a citrus fruit, given the context of the image.
A3) The masked region contains a fruit. It is likely an orange or a tangerine.
Acc: 66.7
S$_{GT}$: 51.9
Con: 66.7
S$_C$: 77.0

**GPT-4V**
A1) I can't identify the object in the masked region of the image. However, the visible parts suggest that the surroundings include plants and possibly some fruit.
A2) I'm unable to see the masked region, so I can't determine what object it might contain. If you can describe the image or provide more context, I may be able to help!
A3) I can't see the masked region in the image you provided, so I'm unable to determine what kind of object is there. If you can describe the object or provide more context, I can help you with information related to it!
Acc: 0
S$_{GT}$: 21.3
Con: 33.3
S$_C$: 58.1

**GPT-4o**
A1) The masked region in the image obscures part of a fruit, specifically an orange, growing on a tree. The surrounding branches and leaves, along with the color and texture of the visible parts, suggest that the object is an orange.
A2) The masked region appears to be obscuring parts of a tree or plant. Given the context of the surrounding branches and leaves, it is likely that the object in the masked region could be fruit or flowers growing on the tree. The background outside the masked area reveals what seems to be some orange-colored fruit, suggesting they could be oranges. Therefore, the object in the masked region is likely a fruit, specifically an orange.
A3) The masked region of the image is likely containing a fruit, most probably an orange or a similar citrus fruit, as suggested by the context of the surrounding branches and leaves. The surrounding foliage and the small visible parts of an orange fruit further hint towards this.
Acc: 100
S$_{GT}$: 47.1
Con: 100
S$_C$: 91.5

Figure 10: Qualitative Results for the **Context Reasoning Task**.

**Prompt**: What kind of object is in the masked region?

**GT**: toilet

**BLIP-2**
A1) a sink
A2) a sink
A3) a sink

Acc: 0
$S_{GT}$: 48.6
Con: 100
$S_C$: 100

**LLaVa 1.5M (Improved)**
A1) toilet
A2) toilet
A3) toilet

Acc: 100
$S_{GT}$: 100
Con: 100
$S_C$: 100

**BLIP-2 (Improved)**
A1) toilet
A2) toilet
A3) toilet

Acc: 100
$S_{GT}$: 100
Con: 100
$S_C$: 100

**LLaVa 1.5M**
A1) The masked region in the image contains a toilet paper dispenser.
A2) The masked region in the image contains a sink.
A3) There is a sink in the masked region.

Acc: 33.3
$S_{GT}$: 41.3
Con: 0
$S_C$: 53.7

**mPlug_owl2**
A1) The masked region in the image contains a picture of a toilet.
A2) In the masked region, there is a sink faucet.
A3) In the masked region of the image, there is a sink.

Acc: 33.3
$S_{GT}$: 48.8
Con: 33.3
$S_C$: 47.1

**moellava**
A1) The masked region contains a white sink.
A2) The masked region contains a toilet.
A3) The masked region contains a toilet.

Acc: 66.7
$S_{GT}$: 51.0
Con: 33.3
$S_C$: 78.3

**Qwen-VL**
A1) The masked region contains a radiator.
A2) The masked region contains a toilet, which is located next to the sink in the bathroom.
A3) The masked region is a white sink, which is a part of a bathroom.

Acc: 33.3
$S_{GT}$: 35.1
Con: 33.3
$S_C$: 59.4

**BLIP-3**
A1) The masked region appears to be a toilet.
A2) The masked region obscures a toilet.
A3) The masked region is over a sink.

Acc: 66.7
$S_{GT}$: 43.9
Con: 66.7
$S_C$: 76.5

**Gemini**
A1) The masked region contains a toilet bowl.
A2) The masked region contains a toilet bowl.
A3) The masked region contains a toilet brush.

Acc: 100
$S_{GT}$: 62.3
Con: 100
$S_C$: 86.4

**GPT-4V**
A1) The masked region seems to be obscuring part of a sink or basin area. It's likely a portion of the bathroom fixtures, possibly where you would find taps or plumbing connections. However, without more specifics, it's hard to determine exact details.
A2) The masked region likely contains an object found in a restroom setting, possibly concerning sanitary fixtures. Based on the surrounding objects visible in the image, it could be a toilet or another type of sink or basin.
A3) The masked region likely contains a washbasin or sink, as suggested by the shape and position of the visible parts of the object in the image.

Acc: 33.3
$S_{GT}$: 27.6
Con: 100
$S_C$: 75.7

**GPT-4o**
A1) The masked region seems to be concealing an object that is directly next to a sink and appears to be part of a restroom setup. Given the context and typical restroom arrangements, it's possible that the masked object is another fixture typically found in such settings, such as a paper towel dispenser, a hand dryer, a trash bin, or part of a urinal. The exact nature of the object cannot be determined solely based on the surrounding context.
A2) The masked region is likely covering a toilet bowl. The objects in the surrounding area include a urinal to the left and a sink in the center, often found together in a restroom environment with a toilet being a common nearby fixture.
A3) The masked region appears to be covering part of a toilet bowl in a bathroom.

Acc: 66.7
$S_{GT}$: 41.0
Con: 100
$S_C$: 85.6

Figure 11: Qualitative Results for the **Context Reasoning Task**.

