# OpenReview forum: "MM-R$^3$: On (In-)Consistency of Multi-modal Large Language Models (MLLMs)"
_ICLR.cc/2025/Conference — Submitted to ICLR 2025_

### Official Review · Reviewer_p3kh · 2024-10-31

**Soundness:** 3
**Presentation:** 3
**Contribution:** 3
**Rating:** 6
**Confidence:** 3

**Summary:**

This paper addresses the often-overlooked aspect of **consistency** in Multi-modal Large Language Models (MLLMs). While most existing evaluations focus on accuracy across various tasks, this work introduces the **MM-R³ benchmark** to assess the consistency of MLLMs when presented with semantically similar inputs. The benchmark consists of three tasks:

1. **Question Rephrasing**: Evaluating consistency in responses to different phrasings of the same question.
2. **Image Restyling**: Assessing consistency when images are presented in different styles.
3. **Context Reasoning**: Testing the model's ability to infer masked or occluded content in images.

The authors analyze several state-of-the-art MLLMs, both open-source and proprietary, and find that consistency does not always align with accuracy. They observe significant variability in consistency across models. To address this, they propose a simple **adapter module** that can be integrated into existing MLLMs to enhance consistency. Experiments demonstrate that this approach leads to notable improvements in consistency metrics without significantly altering accuracy.

**Strengths:**

- **Novel Benchmark**: The MM-R³ benchmark is a valuable contribution, focusing on consistency—an important but underexplored aspect of MLLM evaluation.

- **Comprehensive Analysis**: The paper provides a thorough evaluation of multiple state-of-the-art MLLMs across different tasks, offering insights into their consistency and accuracy.

- **Clear Motivation**: The paper clearly articulates the importance of consistency in AI systems for robustness and trustworthiness.

- **Well-Structured and Clear Presentation**: The paper is well-organized, making it easy to follow the methodology, experiments, and findings.

**Weaknesses:**

- **Lack of Analysis**: The paper does not delve into how different pretraining strategies or model architectures contribute to the observed inconsistencies. An analysis from the pretraining perspective or model architecture could provide deeper insights into why certain models perform better in terms of consistency.

- **Adapter Evaluation**: There should be analysis about the overhead brought by the adapter and models' general performance influenced by the adapter.

- **Lack of Error Analysis**: The paper could benefit from a deeper analysis of failure cases to understand why models are inconsistent and how the adapter mitigates these issues.

- **Lack of Novelty**: The adapter module lack novelty compared to existing methods in the field. The paper does not sufficiently compare or contrast its approach with prior work on improving consistency, which could make the contribution seem incremental.

**Questions:**

1. **Adapter Overhead**: What is overhead brought by the adapter?

2. **Impact on Downstream Tasks**: Does the adapter module impact the models' performance on other downstream tasks?

3. **Error Analysis**: Can you provide more insights into the types of inconsistencies observed in the models and how the adapter addresses them? For example, are there specific patterns or error types that the adapter helps mitigate?

4. **Pretraining Analysis**: Could you provide an analysis of how different pretraining strategies or model architectures impact consistency? Are certain architectures more prone to inconsistency, and if so, why?

---

> ### Author Response · Authors · 2024-11-25
> **Responses to reviewer**
>
> **Adapter Overhead: What is overhead brought by the adapter?**
>
> *Adapter parameters:* BLIP-2 adapter parameters are 376M and LLaVA adapter parameters are 268M. The original number of parameters in BLIP-2 is 12.1B and in LLaVA is 7B. This means that the overhead of the adapter is only 3.1% and 3.8% respectively.
>
> *Inference speed:* We run 100 examples on the original LLaVA and adapter + LLaVA (ours). The processing time for the original LLaVa is 89.4 seconds and ours is 91.2 seconds on a single GPU with batch size = 1.
>
>
> **Impact on Downstream Tasks: Does the adapter module impact the models' performance on other downstream tasks?**
>
> We evaluate the OKVQA dataset to validate the performance of the downstream task before and after the adapter-based fine-tuning of the MLLM model.
>
> - Original LLaVa 1.5M (temperature = 0): Acc = 58.04
> - Finetune LLaVa 1.5M (temperature = 0): Acc = 57.12
>
> The number does not drop after adding the adapter, indicating the proposed adapter can not only preserve the ability of MLLM but also improve the consistency.
>
>
> **Error Analysis: Can you provide more insights into the types of inconsistencies observed in the models and how the adapter addresses them? For example, are there specific patterns or error types that the adapter helps mitigate?**
>
> While it is difficult to quantify specific trends due to variability of tasks and questions. One interesting behaviour that we observe are inconsistencies in numeric responses and how the adapted model is able to address them by inducing consistency. Some examples are below:
>
> ***Example 1:***
>
> gt: 36
>
> original: 250, 10%, 69%
>
> finetune: 50%, 40%, 42%
>
> ***Example 2:***
>
> gt: \$2,114.99
>
> original: no., 1200, \$2698
>
> finetune:
> $2,926,
>
> $2,823,
>
> $2,355
>
>
> **Could you provide an analysis of how different pretraining strategies or model architectures impact consistency?**
>
> We notice that trends with respect to architecture and training strategies depend on types of perturbation that we consider. For perturbations to visual inputs (stylizations), we notice that even though the performance of the BLIP-2 model and LLaVa model are similar in terms of accuracy (Tab3: image restyling) the consistency of the BLIP-2 model is much less compared to the LLaVa model. This points to the fact that models without instruct tuning, such as BLIP-2, are weaker and more susceptible to stylizations-based perturbations to input images and less consistent than LLaVa family counterparts.
>
> We notice from Table 5 that in terms of architecture, scaling the MLLM language decoder to a larger size (13B vs 7B) helps make the model more consistent overall. We attribute this to the fact that using a larger language decoder during the fine-tuning stage of LLM training helps with more effective knowledge transfer between visual and language modality making the models less susceptible to changes to input which leads to more consistency.
>
> On the other hand, we notice that in the case of occluded objects that require reasoning based on the overall semantic context of the scene, the BLIP-2 model is much more consistent than the LLaVa models. We attribute this to the hybrid of multiple losses (image-text matching and image-text contrastive learning) used pretraining of Qformer of Blip-2 which helps them capture the overall semantic context of the image better compared to LLaVa family of models which does not involve pretraining with these losses.

---

> > ### Comment · Reviewer_p3kh · 2024-11-27
> >
> > Thank you for the response! I have no additional questions and will keep my rating.

---

> > > ### Author Response · Authors · 2024-11-28
> > > **Responses to reviewer**
> > >
> > > Thank you for your response, we’re glad to know we addressed all your comments. Kindly let us know if you need any further clarification that might help you increase your score.

---

### Official Review · Reviewer_zYQt · 2024-11-03

**Soundness:** 4
**Presentation:** 4
**Contribution:** 3
**Rating:** 6
**Confidence:** 4

**Summary:**

The paper studies the consistency of MLLMs using 3 tasks: (a) when the question is rephrased with the same meaning; (b) when the image is re-styled; (c) when the image is partially occluded. The contributions include a new dataset, intensive analysis of representative models, and an adapter-based method to improve the consistency (and accuracy). The dataset is constructed using images/questions from existing datasets with modifications generated using LLMs or image generation models. The analysis, including both open MLLMs and private ones like Gemini and GPT-4, reveals that consistency does not necessarily correlate with accuracy. The method shows that the adapter can improve the consistency (with slight improvement in accuracy as well) for LLaVA-1.5 and BLIP-2.

**Strengths:**

1. The problem is clearly defined. The dataset and the method are intuitive.
2. The experiments are intensive, covering a variety of models. The comprehensive analysis revealing that consistency not correlating with accuracy is also interesting.
3. Writing is clear and easy to follow, providing clear details for the experiments.

**Weaknesses:**

1. While the results are intensive, it is a bit overwhelming to look at each of the three tasks and four metrics one by one. Is there a metric that can be a good proxy of all the results, or can the average be a good representative?
2. It is great that the adapter shows clear improvements on the proposed dataset, but it also worths checking the results on standard datasets like VQAv2. After adding the adapter, does the performance on standard datasets drop?
3. This paper is not the first one to study “consistency” in VQA/MLLMs. More discussion and comparisons with existing works should be provided.

**Questions:**

See weakness.

---

> ### Author Response · Authors · 2024-11-25
> **Responses to reviewer**
>
> **While the results are intensive, it is a bit overwhelming to look at each of the three tasks and four metrics one by one. Is there a metric that can be a good proxy of all the results, or can the average be a good representative?**
>
> This is an excellent suggestion. Motivated by works in generalized few-shot recognition, we propose to use the Harmonic mean of correctness and consistency as a single good proxy metric. We first calculate the average of Acc and S_GT, two metrics that evaluate correctness against the ground truth. Next, we compute the average of Con and Sc, two metrics assessing the consistency of generated responses. Finally, we combine these two averages into one single score using the harmonic mean, as we believe this approach can reduce bias when averaging values with large disparities. We use harmonic mean since ideally we want a model to be both correct and consistent and it helps balance the performance between these two key aspects.
>
> Final_score = Harmonic_mean(mean(Acc + SGT),  mean((Con + Sc))
>
> | Harmonic mean | BLIP-2 | mPLUG-Owl2 | LLaVa | MoELLaVa | Qwen-VL-Chat | BLIP-3 | Gemini | GPT-4V | GPT-4o |
> |:---:|:---:|:---:|:---:|:---:|:---:|:---:|:---:|:---:|:---:|
> | Rephrasing | 46.00 | 45.97 | 51.16 | 47.82 | 57.52 | 50.42 | 58.04 | 60.43 | 61.94 |
> | Styling | 23.15 | 18.86 | 21.30 | 24.54 | 18.26 | 23.49 | 23.98 | 21.07 | 27.35 |
> | Masking | 48.10 | 33.39 | 47.75 | 38.31 | 31.57 | 38.38 | 54.87 | 33.86 | 48.57 |
>
> **After adding the adapter, does the performance on standard datasets drop?**
>
> We evaluate the OKVQA dataset to validate the performance of the downstream task before and after the adapter-based fine-tuning of the MLLM model.
>
> - Original LLaVa 1.5M (temperature = 0): Acc = 58.04
> - Finetune LLaVa 1.5M (temperature = 0): Acc = 57.12
>
> The number does not drop after adding the adapter, indicating the proposed adapter can not only preserve the ability of MLLM but also improve the consistency.
>
>
> **This paper is not the first one to study “consistency” in VQA/MLLMs. More discussion and comparisons with existing works should be provided.**
>
> To the best of our knowledge, our paper is the first to do an in-depth study of diverse MLLMs and improve consistency in the realm of multimodal vision language models which integrate a visual encoder with an LLM and give continuous textual captions/responses to queries. We are aware of papers that study this property in LLMs, which we discuss in related work.  We will be happy to provide additional discussions and comparisons if the reviewer can identify specific works with respect to which such discussions and comparisons should be made.

---

> > ### Comment · Reviewer_zYQt · 2024-11-27
> > **Thanks for the rebuttal**
> >
> > I thank the authors for the rebuttal. I suggest add the results into the main paper. Meanwhile, for results on OKVQA, 58.04 -> 57.12 does seem like a small drop, for which more discussion will by helpful.
> >
> > I will keep my score as 6.

---

> > > ### Author Response · Authors · 2024-11-28
> > > **Responses to reviewer**
> > >
> > > Thank you for your response. We will add these results, and all relevant content from the rebuttal to the main paper (space permitably) and appendices. Regarding the OKVQA results, we believe the small gap in performance (58.04 → 57.12, less than 1%) is justifiable given the much larger improvements in consistency resulting from the adapted model. Further, we believe this minor gap can potentially be addressed with more complex techniques (e.g., joint adapter training and distillation from the original LLaVa model). Kindly let us know if you need any further clarification that might help you increase your score.

---

### Official Review · Reviewer_hrqk · 2024-11-03

**Soundness:** 3
**Presentation:** 3
**Contribution:** 3
**Rating:** 6
**Confidence:** 3

**Summary:**

This paper introduces a benchmark to evaluate the accuracy and consistency (with emphasis on consistency) for MLLMs. The benchmark contains the tasks of question rephrasing, image restyling and context reasoning. The benchmark has been utilized to evaluate and compare the SOTA open- and closed-sourced MLLMs, and the results revealed the limitations of most of these models in the consistency and the relationship between the consistency and accuracy. The paper also proposes an adapter to be deployed between the encoder and decoder of a MLLM to improve the consistency. The evaluation results demonstrate the improvements for two of the models, but the resulting consistency is still low.

**Strengths:**

1. The work is well-motivated. The existing benchmarks have focused on model accuracy, but consistency has been overlooked.

2. The paper is well organized and written. In general, the benchmark design, the evaluate methodology, and the result analysis have been elaborated clearly.

3. The proposed adapter has been demonstrated helpful in improving the consistency (mainly) and accuracy of two of the MLLMs.

**Weaknesses:**

1. While 6 open-sourced MLLMs were analyzed for consistency and accuracy, only two of them (BLIP-2 and LLaVa 1.5M) were used in experiment to evaluate the proposed adapter. It is not clear on its effectiveness on other models. Also, the improved consistency, especially for image restyling, is still low compared to other models.

2. Figures 2, 3, and the embedded figures are too small to view clearly.

**Questions:**

The evaluation results show the consistency due to stochasticity of the models. How does the stochasticily affect the measured inconsistency based on rephrasing, restyling and context reasoning tasks? How does the proposed adapter address the consistency caused by the stochasticity?

---

> ### Author Response · Authors · 2024-11-25
> **Responses to reviewer**
>
> **Why choose BLIP-2 and LLaVa?**
>
> As mentioned in Section 5.2 Implementation Details (line 445), we choose BLIP-2 and LLaVa 1.5M for consistency improvement experiments because they are widely used models, have low consistency compared to other models and allow us to show the efficacy of our approach on different types of MLLM families (i.e., ones that use only CLIP v.s Qformer based architectures). These models have also served as the foundation of many newer SoTA open-source MLLM models (such as, BLIP3 and LLaVA-NEXT). Generally, we view our adapter and experiments in the paper as a proof of concept that one can improve consistency of MLLM models without necessarily impacting their accuracy. While we show that the adapter can be used to improve consistency of pre-trained models effectively, ultimately we imagine that consistency objectives would be embedded into RLHF fine-tuning and other mechanisms of training MLLMs in the future.
>
> **Image restyling is still low compared to other models.**
>
> We agree that the image restyling task showed less improvement compared to the question rephrasing and context reasoning tasks. We believe this could be due to the inherent difficulty of the task for MLLMs, which have generally not seen images of this form.
>
> **Figures 2, 3, and the embedded figures are too small to view clearly.**
>
> Thanks for the suggestion. We will enlarge the figures in the revised version as much as space permits.
>
> **How does the stochasticity affect the measured inconsistency based on rephrasing, restyling and context reasoning tasks?**
>
> As shown in Figure 3: Impact of Entropy, consistency in BLIP-2, BLIP-3 and Qwen-VL-chat are less affected by the stochasticity, while LLaVa 1.5, MoE-LLaVa and mPlug-Owl2 are more sensitive to the entropy parameters.
>
> **How does the proposed adapter address the consistency caused by the stochasticity?**
>
> To address this question, we run different entropy on the LLaVa 1.5M model with the proposed adapter. Expectantly, higher temperature does lead to lower accuracy and consistency. However, adapter counterparts perform significantly better in consistency, compared to non-adopted counterparts, for the same temperature. Specifically, we see 10.63 point improvement at temperature of 0.7 and 13.81 improvement at temperature of 0.2 in terms of Con metric; improvement in Sc are similar ~10%.
>
> |  | Acc | S_GT | Con | Sc |
> |---|:---:|:---:|:---:|:---:|
> | LLaVa Original (temp 0.7) | 26.94 | 59.22 | 32.54 | 53.80 |
> | LLaVa Original (temp 0.2) | 31.20 | 62.62 | 45.97 | 62.39 |
> | LLaVa + adapter (temp 0.7) | 31.37 | 65.91 | 43.17 | 62.26 |
> | LLaVa + adapter (temp 0.2) | 34.87 | 68.51 | 59.78 | 72.78 |

---

> ### Comment · Reviewer_hrqk · 2024-12-02
>
> Thank you for answering the questions. I will keep the original rating.

---

### Official Review · Reviewer_L2UA · 2024-11-04

**Soundness:** 2
**Presentation:** 2
**Contribution:** 1
**Rating:** 3
**Confidence:** 4

**Summary:**

This paper presents MM-R3 as a dataset to study the effect of language and image shifts in MLLMs. The language shifts in MM-R3 are designed such that they diversify the input to the LLM stream but retain the original semantics. The images are affected by style shifts and the effects on the MLLM’s are tested with over 13.3K Testing examples. MM-R3 evaluates the accuracy of the replies and also the consistency of the MLLM outputs

**Strengths:**

The paper explores an interesting research direction

**Weaknesses:**

In a benchmark with over 87.000 examples, visually inspecting only 100 random question-rephrasing pairs and 100 images (about 0.002% of the data) is a poor quality metric. Even if 92% language rephrasing and 86% image are semantically equivalent in this extremely small sample, the statistics for the entire dataset could differ strongly.
My main concerns about the MM-R3 benchmark are the lack of novelty and the impracticality of the proposed benchmark.
There is extensive literature analyzing (Empirically and theoretically) the effects of subtle shifts of the Network outputs with respect to the shift on the inputs. Among many others, [A][B][C][E] analyze the robustness of CNNs and Transformers to alterations in their visual input. Other works have already provided the theoretical analysis of the trade off between  accuracy and robustness [D]. Regarding the language modifications the sensitivity towards changes in the input has also been extensively studied [F][G][H]. The joint model is not an exception and theoretical analysis is already available[I]. This paper confirms already known behavior where even small changes in the input can produce alterations on the output, and that training on the shifted data can help reduce the variability of the output. Beyond that, I can not find any other novel insights or results in this paper. Therefore I consider MM-R3 and its results as yet another empirical evaluation inside this line of work.
Regarding the practicality of the MM-R3 benchmark, the Image-restyling tasks seem to be a far fetched task. The authors focus on strong style shifts on their images that are only possible with direct human intervention. I can not imagine any real-world condition or task where a captured image is so distorted  that it resembles the visual patterns in the Candy Mosaic and Udnie styles. Only the gray-scale looks like a reasonable artifact to find in images, but is only a small component of this benchmark. Why do we need to assess that  MLLMs are robust to such exotic visual perturbations?
Following a similar idea, could the authors elaborate on what real-world task or established benchmark requires the MLLM to correctly guess an occluded object?. In addition, If an MLLM guesses correctly, does it make it more accurate/suitable for a given task?, would a user even benefit from having improved results on this task?.
MLLMs can already perform context reasoning  based on spatial locations, direct object relationships, and even image sequences. In comparison the proposed context reasoning represents an Ill posed task where many plausible objects could be hidden behind the occluding shapes. Since context relationships are already a strong component in many current benchmarks[J][K], could the authors elaborate why we need to evaluate Context Reasoning using an Ill posed task.
Regarding the proposed module, Table 6 does not represent a direct and fair evaluation between the baseline (Ori.) and the proposed method (Adapt.). The Adapt. row has been trained for the specific tasks contained in MM-R3, meanwhile the models in Ori have never been trained for them. In other words, The Ori. models are operating in a zero-shot manner, while the Adapt. models are fine-tuned and specialized for the target task. This is a clear disadvantage for the Ori. models which explains the performance gap.
A fair comparison would retrain the original model without using the proposed adapter module and compare if there is any improvement between the finetuned and finetuned+adapter module. For a complete assessment, different architectures for the module could be tested, this will validate that the proposed architecture is optimal for the fine-tuning in MM-R3. FInally the performance in standard benchmarks should be tested once again to assure the capabilities of the model in other tasks and datasets have not been altered.
Without a complete and fair comparison, the adapter module can not be presented as a contribution.
I cannot be certain about the quality of the benchmark, two of the proposed taks look impractical and don't seem to be targeting any useful application case of MLLMs, and the empirical validation is flawed. Therefore I’m recommending rejection.

[A] Carlini, Nicholas, and David Wagner. "Towards evaluating the robustness of neural networks." 2017 ieee symposium on security and privacy (sp). Ieee, 2017.

[B] Croce, Francesco, and Matthias Hein. "Reliable evaluation of adversarial robustness with an ensemble of diverse parameter-free attacks." International conference on machine learning. PMLR, 2020.

[C] Ilyas, Andrew, et al. "Adversarial examples are not bugs, they are features." Advances in neural information processing systems 32 (2019).

[D] Zhang, Hongyang, et al. "Theoretically principled trade-off between robustness and accuracy." International conference on machine learning. PMLR, 2019.

[E] Bhojanapalli, Srinadh, et al. "Understanding robustness of transformers for image classification." Proceedings of the IEEE/CVF international conference on computer vision. 2021.

[F]Jin, Di, et al. "Is bert really robust? a strong baseline for natural language attack on text classification and entailment." Proceedings of the AAAI conference on artificial intelligence. Vol. 34. No. 05. 2020.

[G]Zhu, Kaijie, et al. "Promptbench: Towards evaluating the robustness of large language models on adversarial prompts." arXiv preprint arXiv:2306.04528 (2023).

[H]Moradi, Milad, and Matthias Samwald. "Evaluating the robustness of neural language models to input perturbations." arXiv preprint arXiv:2108.12237 (2021).

[I] Li, Linjie, Zhe Gan, and Jingjing Liu. "A closer look at the robustness of vision-and-language pre-trained models." arXiv preprint arXiv:2012.08673 (2020).

[J]Antol, Stanislaw, et al. "Vqa: Visual question answering." Proceedings of the IEEE international conference on computer vision. 2015.

[K]Liu, Yuan, et al. "Mmbench: Is your multi-modal model an all-around player?." European Conference on Computer Vision. Springer, Cham, 2025.

**Questions:**

Why do the authors choose different source datasets?. The dataset diversity is clearly welcomed, but I wonder why the authors refrain from performing image re-styling over MSCOCO or over the image data of InfographicsVQA. Likewise MSCOCO contains caption data that can be rephrased.
In line 246, when the authors state “the ground truth annotation is encompassed within the MLLM’s response”. What exactly is being tested? This reads as if the authors test for the GT to be a substring of the answer of the MLLM. Could the authors confirm the exact evaluation procedure? How is the text output tested and labeled as Correct or Erroneous?

---

> ### Author Response · Authors · 2024-11-25
> **Responses to reviewer**
>
> **Validation of dataset quality.**
>
> We appreciate the concern raised by the reviewer. We would like to clarify that 100 samples are 0.2% of the dataset and not 0.002% as stated by the reviewer. Nevertheless, the point stands. Since validating a large portion of the dataset (with 87,000 samples) manually would be exceedingly costly and time-consuming, we adopt two alternate strategies to further evaluate the quality of our dataset and address the concern. (1) We human validate additional 200 samples during the rebuttal period and find the statistics (on over 300 random samples now) are not very different (93% semantic equivalency for language rephrasing vs. 92% reported on 100 samples in the paper; 85% semantic equivalence for restyling vs. 86% reported on 100 samples in the paper). These additional results illustrate that the quality metrics reported are stable and reflective of the dataset as a whole. (2) we use the InternVL-26B [1] model (a strong VLM not part of our analysis, with capabilities exceeding GPT4-o in many cases) to automatically validate ALL of the data for the rephrasing task and find it to be 88% semantically equivalent according to InternVL. Note that this is likely a lower bound as InternVL itself is not perfect. However, this further validates the quality of the dataset.
>
> [1] InternVL: Scaling up Vision Foundation Models and Aligning for Generic Visual-Linguistic Tasks, Z. Chen, J. Wu, W. Wang, W. Su, G. Chen, S. Xing, M. Zhong, Q. Zhang, X. Zhu, L. Lu, B. Li, P. Luo, T. Lu, Y. Qiao, J. Dai, CVPR 2024.
>
> **Why exotic visual perturbations? Why require an MLLM to correctly guess an occluded object?**
>
> These tasks are designed to probe the “property” of consistency in MLLM models since humans inherently exhibit this property in their responses. Specifically, our definition of consistency is motivated by human and social psychology and the Cialdini’s Principle of Consistency. The Cialdini’s consistency principle states that people are motivated toward cognitive consistency and will change their attitudes, beliefs, perceptions, and actions to achieve it. In other words, humans in certain experimental settings prefer consistency over more objective measures. We believe that it is important for models to also exhibit consistency in order to operate convincingly in tandem with human users. This would also go a long way towards closing the gap in building trust and MLLM use for, and in, decision making processes. Further, please note that we do NOT require models to guess correctly, we mainly require and measure their ability to respond consistently. In other words, a model that responds incorrectly but the same for the various perturbations will be deemed 100% consistent.
>
> Whether the task is realistic or not, is somewhat irrelevant for our ability to measure this property. For example, our restyling task measures whether models respond consistently when texture cues are removed via stylization. Previous work [1,2] shows that humans are shape-biased (and do not change decisions when texture cues are modified via stylization). Operating similar to humans, i.e., being shape-biased has various downstream benefits like better recognition performance and overall robustness. Similarly, humans can guess masked objects from the context of the scene, and even if the guess is incorrect, they would persist on the chosen answer irrespective of the type of mask. The essence of our task is not to directly solve a downstream problem but a way to probe a property of MLLMs that humans inherently exhibit in their decision-making/responses. In fact, we argue that asking more open-ended questions in an ill-posed task is conducive to enhancing our ability to measure the intrinsic consistency/inconsistency of such models. The key insight is that even for these (perhaps unrealistic) perturbations of an image and ill-posed tasks, a given person will respond consistently, and we should expect MLLM models to do the same. This is the key metric that we measure.
>
> [1] ImageNet-trained CNNs are biased towards texture; increasing shape bias improves accuracy and robustness, R. Geirhos, P. Rubisch, C. Michaelis, M. Bethge, F. Wichmann, W. Brendel, ICLR 2019.
>
> [2] Does enhanced shape bias improve neural network robustness to common corruptions, C. Mummadi, R. Subramaniam, R. Hutmacher, J. Vitay, V. Fischer, J. Metzen, ICLR 2021

---

> ### Author Response · Authors · 2024-11-25
> **Responses to reviewer (cont.)**
>
> **Adversarial robustness vs. consistency.**
>
> We appreciate the relationship pointed out by the reviewer. We are aware of the adversarial robustness literature but did not think it was sufficiently close to be discussed. In retrospect, we agree that we should have discussed it as part of the related work. We will revise the manuscript to add such discussion. That said, there are important differences between adversarial robustness (and works cited by the reviewer) and consistency we study in this paper. Specifically, while there is a great variety of works in adversarial robustness, let us contrast them with our work in terms of their main tenets:
>
> 1. Most adversarial robustness approaches [A, B, C, D, E] operate in classification settings. Models such as [F, G, H] operate on LLMs that have no entitlement of vision and language, and [I] only studies CLIP (which is a particularly simple contrastive VLM variant). Notably, none of the models deal with VLMs with continuous text outputs.
> 2. They assume the presence of an adversary agent that attempts to find small, local, and often imperceptible, perturbations to inputs (e.g., [E] propose pixel level noise perturbations for vision models; [G] propose typos and synonyms for LLMs; [H] propose character and word level deletions, repetition, etc.), that “produce an incorrect response” [G] or a “decrease in overall classification performance” [I]. In other words, robustness is closely tied to accuracy; i.e., robustness only makes sense in the context of a capable model, for samples that the original model is able to classify correctly.
> 3. Adversarial robustness models, particularly those that attempt to provide theoretic guarantees, quantify worst-case performance under an adversary attack.
>
> In contrast, in studying consistency in VLMs we:
>
> 1. Focus on a broad class of VLM models that produce open-world textual outputs (including both open- and closed-sourced); this is well beyond CLIP discussed in [I] (which is the closest among suggested citations).
> 2. We focus on semantic input perturbations (rephrasing and restyling) of both visual and lingual modalities and semantic output equivalence. This is much harder to achieve and quantify. This also goes significantly beyond local word/character perturbations in LLMs or pixel noise perturbations in vision robustness literature.
>
>     Importantly, the notion of consistency is entirely devoid of the accuracy or correctness of the original model. Specifically, we study consistency for both all responses and specifically failure cases (see supplementals). A model can be trivially consistent by always responding with the same phrase, irrespective of the input, however, such a model would not generally be considered either accurate or robust under most standard definitions of those two properties. Further, consistency does not assume an adversary, but rather a cooperative agent. In other words, the only perturbations we consider are those likely to be generated by a “typical” user (not one that tries to fool a model). Overall, consistency does not guarantee robustness.
>
>     On the other hand, a robust model may also not necessarily guarantee consistency, because typical robustness measures ability for an adversary to flip the decision from correct to incorrect. In more complex tasks (e.g., VQA, captioning), there may be multiple correct answers and also many ways to be incorrect. Consistency measures semantic equivalency even within these classes, which robustness typically does not.
>
> 3. Finally, consistency as we define it, is a measure of average performance under semantic perturbation, not one of worst-case performance.
>
> We will elaborate on these connections and differences in the revised manuscript.

---

> ### Author Response · Authors · 2024-11-25
> **Responses to reviewer (cont.)**
>
> **Fairness of comparison.**
>
> For a fair comparison, we train our adapter only on the *original unmodified* OKVQA + InfographicVQA dataset (for rephrasing task); and the *original unmodified* Indoor Scene + Google Landmarks Dataset v2 (for the image restyling task). We call this variant (task adapted). Note that (task adapted) and (consistency adapted – reported in the paper) models have identical architectural structure and are both optimized. In other words, the comparison is no longer with zero-shot (original). Due to the time limit, we conducted the experiment using the LLaVa model only. Nevertheless the trends are clear, while training with task-specific data increases the accuracy on the task (an expected behavior), it does very little to improve the consistency (see Con measure). Training the adapter with rephrasing and restyling data substantially improves the consistency, while not negatively impacting, and typically marginally improving, the accuracy. Specifically the improvement in consistency is **33.2 -> 43.2** and **22.5 -> 32.6**, over 10 points for each of the tasks. The lack and quantification of consistency in original models and improvement of consistency through training of the adapter are our key contributions.
>
> |  | Acc | S_GT | Con | Sc |
> |:---:|:---:|:---:|:---:|:---:|
> | Question Rephrasing (original)             | 26.9 | 59.2 | 32.5 | 53.8 |
> | Question Rephrasing (task adapted)         | 28.1 | 66.9 | 33.2 | 56.9 |
> | Question Rephrasing (consistency adapted)  | 31.4 | 65.9 | 43.2 | 62.3 |
> | Image Restyle (original)            | 9.6  | 14.9 | 19.0 | 56.9 |
> | Image Restyle (task adapted)        | 17.3 | 25.7 | 22.5 | 53.3 |
> | Image Restyle (consistency adapted) | 18.1 | 28.1 | 32.6 | 52.6 |
>
>
> **Why choose different source datasets?**
>
> The choice of datasets stems from our desire to maintain diversity in our benchmark, as correctly pointed out by the reviewer, as well as availability of annotations in various source datasets. MSCOCO provides instance segmentation masks, making it ideal for context reasoning task (which requires object masking). MSCOCO itself does not include question-answer pairs (which would be needed for the rephrasing task) or clear scenes to recognize (which is how we organize the restyling task). Some of the MSCOCO images do appear in the VQAv2 dataset, so questions could potentially be obtained from there. We have done some preliminary experiments using such data during the rebuttal, and results are consistent with those obtained on the proposed dataset. Mainly, the relative ranking of models is nearly identical for both accuracy and consistency. The overall accuracy, however, tends to be considerably higher for MSCOCO images (e.g., for Qwen-VL-Chat by as much as 22 to 30 points for rephrasing and restyling), showing that our originally chosen dataset is actually a lot more challenging.
>
>
> **Evaluation of the downstream task.**
>
> We evaluate the original unmodified OKVQA dataset to validate the performance on the downstream task before and after the adapter-based fine-tuning of the MLLM model.
>
> - Original LLaVa 1.5M (temperature = 0): Acc = 58.04
> - Finetune LLaVa 1.5M (temperature = 0): Acc = 57.12
>
> The number does not drop after adding the adapter, indicating the proposed adapter can not only preserve the ability of MLLM but also improve the consistency.
>
> **Evaluation procedure.**
>
> 1. Calculation of Accuracy (Acc):  To clarify line 244, we do case-insensitive substring matching to validate the response. This works because GT responses tend to be single words or short phrases. Consider the example in Figure 5, the answers in the question rephrasing task from LLaVa are “columbia”, “north face”, and “no” and the ground truth answer is “north face”. Hence, Acc for three answers is 0/100/0. As a result, the average score for this example will be 33.3 as reported.
> 2. Calculation of similarity with GT (S_GT): As the exact match criterion has some limitations, i.e. it may inaccurately categorize semantically similar responses as incorrect, we use a similarity metric in the form of Sentence BERT embeddings.
> 3. Calculation of ​​Consistency Accuracy (Con): We compute the pairwise similarity scores between responses using Sentence BERT and utilize a threshold of 0.7 to delineate semantic consistency. Consider again the example in Figure 5, the answers in the question rephrasing task from LLaVa are “columbia”, “north face”, and “no”. Since none of these are semantically similar to one another, the pair-wise Sentence BERT scores are 0.27/0.14/0.24 — all below 0.7 threshold and resulting in Con of 0.
> 4. Calculation of Consistency Similarity (SC): We compute the pairwise similarity scores between responses and average them. Using the same example above, the SC score will be (0.27+0.14+0.24)/3 = 0.21.

---

> > ### Author Response · Authors · 2024-11-28
> > **Responses to reviewer**
> >
> > Thank you again for your comments. We believe we addressed them thoroughly and completely in our rebuttal. We would appreciate it if you would take a look at our responses and let us know if they address the concerns raised in the original review. If you have further questions, comments or concerns, we would gladly address them in the rebuttal time that remains.

---

### Author Response · Authors · 2024-11-25
**Responses to all reviewers**

We thank the reviewers for their valuable feedback and acknowledge that the problem is well-defined and motivated, the proposed dataset is novel, the paper is well-written and organized, the insights are interesting and crucial, the comprehensive analysis reveals important findings and the proposed adapter demonstrates helpful improvements. We individually address reviewer concerns in our responses.

---

### Meta-Review · Area_Chair_nQxk · 2024-12-18

**Metareview:**

This paper studies the consistency issue of the MLLM. Concretely, when prompting semantic similar questions, whether the model could output similar / consistent responses. To study this task, the paper proposed the MM-R$^3$ dataset which contains three tasks: question rephrasing, image restyling, and context reasoning. The paper also proposed an adapter based approach to improve the MLLM model consistency.

Strength:
1. The paper studies a quite interesting and important task: consistency in the MLLM.
2. The paper proposed a dataset that might potentially be useful for consistent study.

Weakness:
1. Only a small portion of the dataset is verified by human for the quality. Although the results suggest the dataset has high quality, it is not guaranteed that the whole dataset is consistent in terms the quality measurement.
2. [More important] The questions in the dataset might not have only one reasonable answer. This means the model might be able to answer in several ways. They are all viable answers. (e.g the examples in Page 10 and the page 26) Especially for those two examples, a lot of answers might be possible and correct. If the dataset contains many of those questions, I am not fully convinced that enforcing the consistency across the response for those questions is essential and important.

Final Decision: Reject

Reason for the final decision:
I am fully supportive for studying the consistency issue in the MLLM. However, we might need to first categorize which kind of questions should be answered in a consistent way. Disagreeing to reviewer L2UA, the image restyling is a good task for study the consistency. Because we would expect the model to response in a consistent way for what is in the image / where is the photo, etc. Another good example is to answer the math questions. If we prompt the model in different way for the same math question, we would expect the model to response in a consistent way. However, for context reasoning and question rephrasing, I am not very sure whether we need the model to answer consistently. Let's take a more extreme case that the user prompt question: "Hey, what's up." We don't hope the model to always respond as "Hey" or "Hi".
I think this paper is a good start for the consistent study. I would suggest the author to first study what kind of tasks need the model to respond consistently. And then updating the paper to reflect those tasks.

**Additional Comments On Reviewer Discussion:**

The major arguments are b/w the reviewer L2UA and the author. The reviewer mainly arguing four points: 1. data quality, 2. fairness in comparison b/w adapter trained model and the original model, 3. the validity of the task (context reasoning), 4. novelty.
All the other reviewers recommend borderline accept.

I think the author responses somehow alleviate the concern in 1. data quality. Addressed concern in 2. fairness. Fully addressed in 4. novelty. However I think the author didn't address point 3.

Given this paper is more about presenting an evaluation benchmark, point 3 is an important question needs to be addressed. However addressing point 3 would require a major update of this paper, which leads to the final decision.

---

### Decision · Program_Chairs · 2025-01-22

Reject